



# Growth rate rather than temperature affects the B/Ca ratio in the calcareous red alga *Lithothamnion corallioides*

Giulia Piazza[1], Valentina A. Bracchi[1], Antonio Langone[2], Agostino N. Meroni[3], Daniela Basso[1]

[1] Department of Earth and Environmental Sciences, University of Milano-Bicocca, Piazza della Scienza 4, 20126 Milano, Italy
[2] CNR – Institute of Geosciences and Earth Resources, Via Ferrata 1, 27100 Pavia, Italy
[3] Department of Civil and Environmental Engineering, Politecnico of Milan, Piazza Leonardo da Vinci 32, 20133 Milano, Italy

*Correspondence to*: Giulia Piazza (g.piazza15@campus.unimib.it)

**Abstract.** The B/Ca ratio in calcareous marine species is informative of past seawater $CO_3^{2-}$ concentrations, but scarce data exist on B/Ca in coralline algae (CA). Recent studies suggest influences of temperature and growth rates on B/Ca, the effect of which could be critical for the reconstructions of surface ocean pH and atmospheric $pCO_2$. In this paper, we present the first LA-ICP-MS analyses of Mg, Sr, Li and B in the CA *Lithothamnion corallioides* collected from different geographic settings and depths across the Mediterranean Sea and in the Atlantic Ocean. We produced the first data on temperature proxies (Mg, Li and Sr/Ca) and B/Ca in a CA species grown in different Basins (the Mediterranean Sea and the Atlantic Ocean), from shallow to deep waters (12 m, 40 m, 45 m and 66 m depth). We tested the B/Ca correlation with temperature proxies and growth rates, in order to evaluate their possible effect on B incorporation. Our results showed a growth rate influence on B/Ca, especially in the deepest sample (Pontian Isl., Italy; 66 m) and in the shallowest sample (Morlaix, Atlantic coast of France; 12 m), where the growth rates were respectively 0.11 mm/yr and 0.13 mm/yr and the B/Ca was respectively 462.8 ± 49.2 μmol/mol and 726.9 ± 102.8 μmol/mol. A positive correlation between B/Ca and the temperature proxies was found only in Morlaix, where the seasonal temperature variation (ΔT) was the highest (8.90 °C). These pieces of evidence suggest that growth rates, triggered by the different ΔT and light availability across depth, affect the B incorporation in *L. corallioides*.

## 1 Introduction

Trace element variations in marine calcareous species inform the reconstruction of changes in the environmental parameters, which characterized the seawater during their growth (Hetzinger et al., 2011). Boron is incorporated into the mineral lattice of calcareous marine species during calcite precipitation. In the ocean, B occurs in two molecular species: boric acid $B(OH)_3$ and borate ion $B(OH)_4^-$ (Dickson, 1990), which are related by the following acid-base equilibrium reaction:

$$B(OH)_3 + H_2O \leftrightarrow B(OH)_4^- + H^+ \qquad \text{Eq. (1)}$$



that shows the dependence of the two species concentration on pH. The equilibrium constant of the reaction $K_B$ is $10^{-8.6}$
which is equal to a $pK_B$ of 8.6 (Dickson, 1990), very similar to seawater pH (~ 8) (Zeebe and Wolf-Gladrow, 2001). Hence,
in normal pH conditions, the reaction described in Eq. (1) is at equilibrium. As the pH lowers, based on Eq. (1), the $[H^+]$ in
seawater increases together with the $[B(OH)_3]$, while the $[B(OH)_4^-]$ decreases. Boron has two stable isotopes: the heavier $^{11}B$
(80.1% of the total abundance) and the lighter $^{10}B$ (19.9%) with an enrichment factor $\varepsilon^{11}B$ ($\delta^{11}B(OH)_3 - \delta^{11}B(OH)_4^-$) equal to

$27.2 \pm 0.6$‰ (Klochko et al., 2006). Seawater isotopic composition $\delta^{11}B_{sw}$ is 39.61‰ (Foster et al., 2010) and varies with the
isotopic composition of $B(OH)_3$ and $B(OH)_4^-$, both enriched in $\delta^{11}B$ with increasing pH (Dickson, 1990). The first analyses
of the isotopic signal of marine carbonates evidenced a strong similarity with the isotopic composition of $B(OH)_4^-$ in
solution, suggesting that borate would preferentially be incorporated into marine carbonates (Vengosh et al., 1991; Hemming
and Hanson, 1992). Therefore, the $\delta^{11}B$ of marine carbonates should increase with pH according to the $\delta^{11}B$ of $B(OH)_4^-$.

The $pCO_2$, the seawater $[CO_3^{2-}]$, pH and $[B(OH)_4^-]$ are mainly controlled by the balance between the total alkalinity (TA)
and the dissolved inorganic carbon (DIC) and are closely related to the $\delta^{11}B$ of the borate (Zeebe and Wolf-Gladrow, 2001;
Dickson et al., 2007). Recent MAS NMR results, though, revealed that 30-48% of $B(OH)_3$ is incorporated in corals,
foraminifera, and calcareous red algae (Klochko et al., 2009; Rollion-Bard et al., 2011; Cusack et al., 2015). The amount of
incorporated boric acid seems to be unrelated to the boron species concentrations in seawater (Cusack et al., 2015) or pH

(Mavromatis et al., 2015) and could be the consequence of a coordination change during biomineralization (Klochko et al.,
2009; Mavromatis et al., 2015).

The above-described theoretical overview frames the role of the B content and its isotopic signature ($\delta^{11}B$) in calcareous
marine species, as recorders of information about the seawater carbonate system. The $\delta^{11}B$ is used to reconstruct past
seawater pH (Hönisch and Hemming, 2005; Foster, 2008; Douville et al., 2010; Paris et al., 2010; Rae et al., 2011) and the

boron-to-calcium ratio (B/Ca) is informative about past seawater $CO_3^{2-}$ concentrations (Yu and Elderfield, 2007; Yu et al.,
2007; Rae et al., 2011).

Warming and acidification are major anthropogenic perturbations of present-day oceans (Alverson et al., 2001; Feely et al.,
2009; Le Quéré et al., 2009; Hansen et al., 2010; Hönisch et al., 2012; Ciais et al., 2013). Ocean acidification reduces the
saturation state of calcite and aragonite, lowering the dissolution threshold of biominerals and threatening habitat-forming

species of critical ecological importance such as coralline red algae (CA) and corals (Morse et al., 2006; Hoegh-Guldberg et
al., 2007; Andersson et al., 2008, 2011; Basso, 2012; Ragazzola et al., 2012; Ries et al., 2016). CA, which precipitate high
Mg-calcite (>8-12 mol% $MgCO_3$) (Morse et al., 2006), are particularly suitable as proxy archives for paleoclimate
reconstruction because of their worldwide distribution and their longevity by indeterminate growth, with no ontogenetic
trend (Adey, 1965; Frantz et al., 2005; Halfar et al., 2008). Moreover, CA thin sections under optical microscopy reveal

bands that reflect the growth pattern (Cabioch, 1966; Basso, 1995a, b; Foster, 2001), similarly to tree rings (Ragazzola et al.,
2016) that can be targeted for high-resolution geochemical analyses. Seasonal growth bands, indeed, consist of the
perithallial alternation of dark and light bands that together constitute the annual growth patterns (Freiwald and Henrich,
1994; Basso, 1995a, b; Kamenos et al., 2009). Dark bands correspond to slow-growing cells produced in the cold season,





which are shorter, thick-walled and with lower Mg contents; while light bands are fast-growing cells produced in the warm
season, which are longer, less calcified and with higher Mg concentrations (Kamenos et al., 2009; Ragazzola et al., 2016).

Most of the literature on boron studies is focused on its isotopic composition (Hemming and Hönisch, 2007; Klochko et al.,
2009; Henehan et al., 2013; Fietzke et al., 2015; Cornwall et al., 2017; Ragazzola et al., 2020), whereas less attention has
been given to B/Ca records, especially in CA. Recent studies suggest that B/Ca is function of seawater pH, as well as of
other environmental variables such as temperature, whose effect should be considered in the attempt to reconstruct surface
ocean pH and atmospheric $pCO_2$ (Wara et al., 2003; Allen et al., 2012; Kaczmarek et al., 2016). Achieving the best
reliability of geochemical proxies for climate reconstructions is indeed crucial, which drives a growing interest on multiple
approaches, by considering multi-proxies for a single environmental factor (D'Olivo et al., 2018; Zinke et al., 2019; Cuny-
Guirrec et al., 2019), as well as the influence of multi-factors on a single proxy (Kaczmarek et al., 2016; Donald et al., 2017).
Actually, the effects of temperature and growth rate on B incorporation have been recently investigated through experiments
on both synthetic and biogenic carbonates (Wara et al., 2003; Yu et al., 2007; Gabitov et al., 2014; Mavromatis et al., 2015;
Uchikawa et al., 2015; Kaczmarek et al., 2016; Donald et al., 2017), showing positive correlations. In particular, B/Ca
experiment on the cultured CA *Neogoniolithon* sp. (Donald et al., 2017), showed positive correlation of B with growth rate,
and a negative correlation with Sr/Ca, which was proposed by the authors as proxy of DIC. Moreover, a cultured experiment
on the high latitude species *Clathromorphum compactum* (Kjellman) Foslie 1898 revealed non-significant temperature
influences on B/Ca and a significant inverse relationship with growth rate (Anagnostou et al., 2019). The current knowledge
on the factors which influence the B incorporation in CA is therefore still controversial.

Nevertheless, no studies have been conducted so far on the correlation between temperature proxies (Mg, Sr, Li /Ca) and
B/Ca. The Mg/Ca ratio is extensively used as a sea surface temperature proxy (SST) in calcifying species, including CA
(Barker et al., 2005; Halfar et al., 2008; Kamenos et al., 2008; Fietzke et al., 2015; Ragazzola et al., 2020), since the
substitution of $Mg^{2+}$ with $Ca^{2+}$ ions in the calcite lattice is an endothermic reaction. Accordingly, Mg incorporation increases
with temperature (Moberly, 1968; Berner, 1975; Ries, 2006; Caragnano et al., 2017). Sr/Ca and Li/Ca ratios in CA have also
been investigated as climate proxies, showing significant positive correlations with temperature in different species, e.g.
*Lithothamnion* spp. (Kamenos et al., 2008; Hetzinger et al., 2011; Caragnano et al., 2014; Darrenougue et al., 2014).

Laser ablation inductively coupled plasma mass spectrometry (LA-ICP-MS) allows high-resolution analysis of a broad range
of trace elements in solid-state samples. This technique has been widely used in biogenic carbonates to extract records of
seawater temperature, salinity and water chemistry (Schöne et al., 2005; Corrège, 2006; Hetzinger et al., 2009, 2011; Fietzke
et al., 2015; Ragazzola et al., 2020). In this paper, we measure by LA-ICP-MS for the first time the temperature proxies
(Mg/Ca, Sr/Ca, Li/Ca) and B/Ca in the non-geniculated CA *Lithothamnion corallioides* (P. Crouan & H. Crouan) P. Crouan
& H. Crouan 1867 collected from different geographic settings and depths across the Mediterranean Sea and in the Atlantic
Ocean. *L. corallioides* is widely distributed in the Mediterranean Sea and in the north-eastern Atlantic Ocean, from Scotland
to Canary Islands (Irvine and Chamberlain, 1999; Wilson et al., 2004; Carro et al., 2014), usually constituting maerl beds
(Potin et al., 1990; Foster, 2001; Martin et al., 2006; Savini et al., 2012; Basso et al., 2017). Here, it forms rhodoliths as





unattached branches (Basso et al., 2016) with obvious banding in longitudinal sections (Basso, 1995b). These characteristics combine to make this species a suitable model for the measurement of geochemical proxies, comparing different

environmental settings. In this paper, we test the B/Ca ratio versus the temperature proxies and the growth rates in order to evaluate their effects on B incorporation, which, indeed, could distort the B signal used for paleoclimate reconstructions.

## 2 Materials and methods

### 2.1 Sampling sites and collection of *Lithothamnion corallioides*

Samples of the CA *L. corallioides* were collected in the Western Mediterranean Sea and in the Atlantic Ocean (Fig. 1). In the

Mediterranean Sea, the samples collected in Pontian Islands (Italy) at 66 m depth were gathered by grab during the cruises of the R/V Minerva Uno, in the framework of the Marine Strategy Campaigns 2017 (Table 1). The last two Mediterranean samples were collected by one of the authors (DB) by SCUBA diving during local surveys at 45 m off the coasts of Pomonte (Elba Island, Italy) (Basso and Brusoni, 2004) and at 40 m depth in the Aegadian Islands (Marettimo, Italy). The Atlantic sample was collected by grab at 12 m depth in Morlaix Bay (Brittany, France) (Table 1).

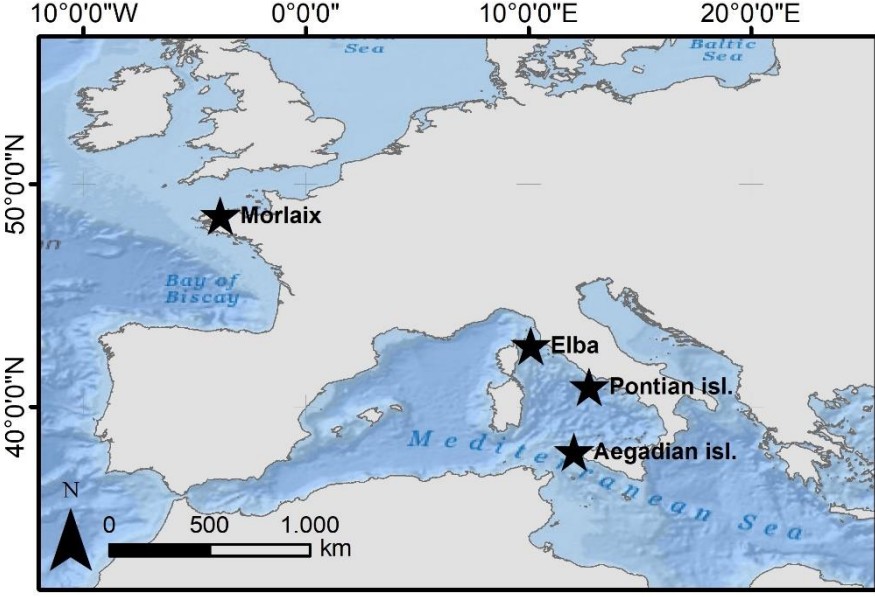


**Figure 1: Map showing the distribution of sampling sites where *Lithothamnion corallioides* samples were collected. Service layer credits: Source Esri, GEBCO, NOAA, National Geographic, Garmin, geonames.org and other contributors.**

Table 1: *Lithothamnion corallioides* samples used in trace elements analyses.

| Sampling site | Longitude/Latitude | Water depth (m) | Collection date |
|---|---|---|---|
| Aegadian Isl. | 37°97'36"N, 12°14'12"E | 40 | 25/08/1993 |
| Elba | 42°44'56.4"N, 10°07'08.4"E | 45 | 01/12/1990 |





| Morlaix | 48°34'42"N, 3°49'36"W | 12 | 02/05/1991 |
| Pontian Isl. | 40°54'N, 12°45'E | 66 | 14/07/2017 |

The identification of the algal samples was achieved by morphological analyses using a Field Emission Gun Scanning

Electron Microscope (SEM-FEG) Geomini 500 Zeiss. Samples were prepared for SEM according to Basso (1995a). Morphological identification was based on Adey & McKibbin (1970), Irvine & Chamberlain (1994), and other information about maerl species distribution provided by Carro et al. (2014) and Melbourne et al. (2017). The samples selection started from a much wider collection than the one eventually selected for the chemical analyses. Particularly, the Atlantic sample (Morlaix) was used as voucher specimen for the subsequent identification of the Mediterranean samples, since

*Phymatolithon* spp. and *L. corallioides* are the major components in the Atlantic maerl (Hall-Spencer et al. 2010; Carro et al., 2014). Hence, once excluded the belonging to the genus *Phymatolithon*, the Morlaix sample identified as *L. corallioides* was used as a reference for the most reliable identification of the other Mediterranean samples.

## 2.2 Sample preparation

The selected algal branches were included in Epo-Fix resin, which was stirred for 2 minutes with a hardener (13% w/w); the

treated samples were then kept drying at room temperature for 24 hours. Afterwards, the included branches were cut by an IsoMet diamond wafering blade 15HC along the direction of growth. In the laboratory of the Institute of Geosciences and Earth Resources (IGG) of the National Research Council (CNR) in Pavia (Italy), the sections were polished with a MetaServ Grinder-Polisher (400 RPM) using a diamond paste solution, finally cleaned ultrasonically in distilled water for 10 minutes and dried at 30°C for 24 hours.

## 130 2.3 Trace elements analyses and environmental data

LA-ICP-MS analyses were carried out at the IGG-CNR laboratory of Pavia (Italy). $^{43}$Ca, $^{7}$Li, $^{25}$Mg, $^{88}$Sr and $^{11}$B contents were measured using an Agilent ICP-QQQ 8900 quadrupole ICP-MS coupled to an Excimer laser ablation system (193 nm wavelength). Element/Ca ratios were calculated from the above cited isotopes, in agreement with Yu et al., (2005) and Darrenougue et al., (2014). Measurements were performed with laser energy densities of 4 J/cm$^2$ and helium as carrier gas.

The laser transect crossed the algal growth direction with a spot size of 50 μm, targeting each growth band change which marked the transition between the cells usually produced in the warm season and those usually produced in the cold season, hereinafter referred to as long and short cells (Fig. 2). Each analysis was carried out in MS/MS mode for 3 minutes by acquiring 60 seconds of background before and after the sampling period by the laser on the polished surface. NIST 612 was used as an external standard whereas Ca was adopted as an internal standard. The Glitter software (v. 4.4.4) was used for

data reduction.



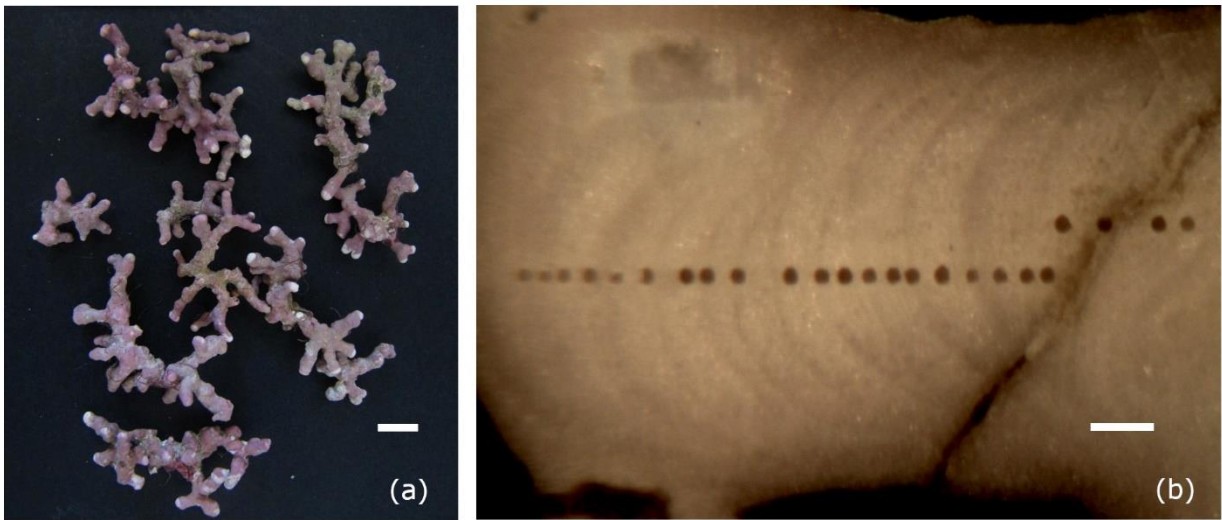

**Figure 2: (a) Thalli of *Lithothamnion corallioides* collected in Morlaix (scale bar = 5 mm). (b) Longitudinal section through the *L. corallioides* branch sampled in Morlaix showing the LA-ICP-MS transects targeting each growth bands (scale bar = 200 µm).**

In the absence of in-situ environmental data, the seawater temperature data have been extracted by at least 11 years of
monthly reanalysis spanning 1980-2017 from ORAS5 (Ocean ReAnalysis System 5), at 0.25-degree horizontal resolution
(Zuo et al., 2019). The nearest sea point of the three-dimensional numerical grid was considered for each sample location.
Details of the time interval considered for each sampling site are shown in Table 2. Minimum, maximum and mean values,
as reported in Table 2, refer to the temperature at sampling depth and have been measured on the entire time interval for
which the data have been extracted.

Carbon data in each sampling site have also been extracted. They were not available in the same time interval of temperature
data. Nevertheless, the seasonal variations occurring in the extracted period have allowed the characterization of the
sampling sites. Monthly mean seawater pH has been derived by the CMEMS (E.U. Copernicus Marine Service Information)
global biogeochemical hindcast spanning 1993-2018, at 0.25-degree horizontal resolution. Monthly means of DIC in 2019
and 2020 have been extracted by CMEMS biogeochemical analysis and forecasts for the Mediterranean Sea, at 0.042-degree
horizontal resolution (Salon et al., 2019; Bolzon et al., 2020). In the Atlantic site, monthly means of DIC were derived from
CMEMS IBI biogeochemical forecasts, at 0.028-degree horizontal resolution covering the years 2019-2020. Minimum,
maximum and mean values of DIC, as reported in Table 2, refer to sampling depth and have been measured on the entire
time interval of extraction.

**2.4 Growth rate estimation**

Growth rates were estimated by microscopical analysis, measuring the length of the LA-ICP-MS transect and dividing it by
the number of annual growth bands. The obtained values, expressed in linear extension over year (mm/yr), were cross-
referenced with Mg/Ca results in order to check for the correspondence of Mg peaks with growth bands. This step was





helpful in highlighting faint bands and to achieve a more reliable estimate of the algal growth. Short cells were referred to slow-growing cells in dark bands, usually produced in the cold season; long cells corresponded to fast-growing cells in light
bands, usually produced in the warm season (Kamenos et al., 2009; Ragazzola et al., 2016).

**2.5 Data elaboration and statistical analyses**

Statistics were performed on both the dataset with all the spot analyses and the dataset with the records from long and short cells separately.

For each spot, a distinction between long and short cells was made by image analyses meant to distinguish the dark growth
bands from the light ones and comparing the results with those obtained by the Mg/Ca ratio measurement.

The Spearman's correlation was tested to provide the statistical comparisons between Mg/Ca, Li/Ca, Sr/Ca and B/Ca records from the LA-ICP-MS analyses in *L. corallioides*. The Kruskal-Wallis test followed by the Dunn's test for comparisons and the One-way ANOVA followed by the Tukey's test for post-hoc analysis were used to compare the geochemical signals among sampling sites and to evidence the differences between group medians and means. All statistical analyses were run in
R 3.6.3 software.

**3 Results**

**3.1 Environmental data**

The temperature data obtained by ORAS5 reanalysis revealed a lower temperature seasonal excursion in the Mediterranean samples with respect to the Atlantic one, as shown by the standard deviation and ΔT values in Table 2. This difference is
explained in terms of the different sampling depths, with the seasonal variations decreasing with increasing depth.

Temperature variations in the Atlantic Ocean were higher, estimated at 8.90 °C in Morlaix bay (France) (minimum monthly mean 8.28 °C and maximum 17.18 °C), registering overall a mean seawater temperature of 12.34 °C (Table 2). Among Mediterranean samples, mean seawater temperatures were highest in the Aegadian Isl. (16.18 °C), followed by the Elba (14.99 °C) and the Pontian Isl. (14.89 °C) (Table 2). Aegadian Isl. also registered the highest temperature variations among
the Mediterranean sites (6.07 °C), with the lowest monthly mean of 13.57 °C and the highest of 19.64 °C (Table 2). Moderate temperature variations characterized the site in Elba (4.61 °C), which registered the lowest monthly mean temperature among Mediterranean sites (12.88 °C) and the maximum temperature of 17.49 °C (Table 2). At the Pontian Isl., consistently with the fact that it is the deepest sampling site at 66 m depth, the lowest seawater temperature variations were found (3.37 °C), with minimum values of 13.34 °C and maximum of 16.70 °C (Table 2). The pH had higher mean values in
Pontian Isl. and Elba (8.13), decreasing in Aegadian Isl. (8.11) and reaching the lowest value in the Atlantic Ocean (Morlaix; 8.06) (Table 2). Similarly, DIC was higher in Pontian Isl. and Elba (2.32 mol/m³), followed by Aegadian Isl. (2.29 mol/m³) and eventually Morlaix (2.17 mol/m³) (Table 2).





**Table 2: (a) Seawater temperature records in each sampling site. The minimum and maximum monthly means of temperature are indicated, as well as the highest temperature variation (ΔT), the mean and the standard deviation of the time series. Data elaborated using the monthly means data extracted by the ORAS5 reanalysis for the time intervals indicated in the last column. (b) pH and DIC data in each sampling site. The minimum, maximum, mean and standard deviation values measured on the time interval 2019-2020 are indicated. Data extracted from monthly means biogeochemical data provided by CMEMS.**

| (a) | Sampling site | Temperature (°C) | | | | | Time interval |
|---|---|---|---|---|---|---|---|
| | | min | max | ΔT | mean | st. dev. | |
| | Aegadian Isl. | 13.57 | 19.64 | 6.07 | 16.18 | 1.38 | 1986-2017 |
| | Elba | 12.88 | 17.49 | 4.61 | 14.99 | 1.21 | 1980-1991 |
| | Morlaix | 8.28 | 17.18 | 8.90 | 12.34 | 2.46 | 1980-1992 |
| | Pontian Isl. | 13.34 | 16.70 | 3.37 | 14.89 | 0.62 | 1986-2017 |

| (b) | Sampling site | pH | | DIC (mol/m³) | |
|---|---|---|---|---|---|
| | | mean | st. dev. | mean | st. dev. |
| | Aegadian Isl. | 8.11 | 0.02 | 2.29 | 0.01 |
| | Elba | 8.13 | 0.02 | 2.32 | 0.01 |
| | Morlaix | 8.06 | 0.04 | 2.17 | 0.01 |
| | Pontian Isl. | 8.13 | 0.01 | 2.32 | 0.00 |

### 3.2 Mg/Ca, Li/Ca, Sr/Ca

Both Li/Ca and Sr/Ca records had positive correlations with Mg/Ca in our samples of *L. corallioides* (respectively r=0.68, p<<0.001 and r=0.64, p<<0.001) (Fig. 3).

The overall mean Mg/Ca was 225.3 ± 30.4 mmol/mol, registering the minimum value in the sample from Aegadian Isl. (171.7 mmol/mol) and the maximum value in Morlaix (311.2 mmol/mol) (Fig. 4).

The Kruskal-Wallis test did not show significant differences in Mg/Ca among samples (p>0.05; Table A1, Fig. 4). Among

Mediterranean sites, the algal sample coming from Aegadian Isl. had the highest Mg/Ca mean value (224.9 ± 30.3 mmol/mol), followed by Elba (223.4 ± 26.4 mmol/mol) and Pontian Isl., which had the lowest Mg/Ca mean value of all sampling sites (216.1 ± 21.9 mmol/mol) (Fig. 4). The highest mean Mg/Ca was registered in the sample from Morlaix bay (239.5 ± 41.2 mmol/mol), which also showed the widest oscillation of Mg/Ca values (Fig. 4).





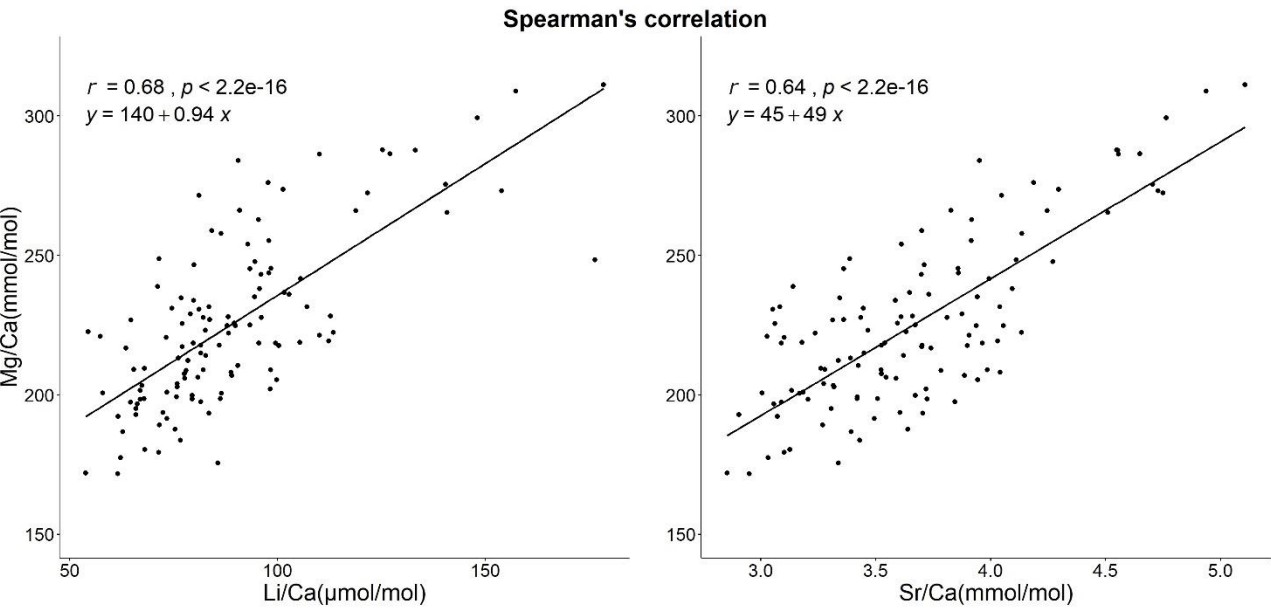

Fig. 3. Correlation plots of Mg/Ca with Li/Ca and Sr/Ca. For each analyses the Spearman's coefficient r, the p-value and the line equation are given.

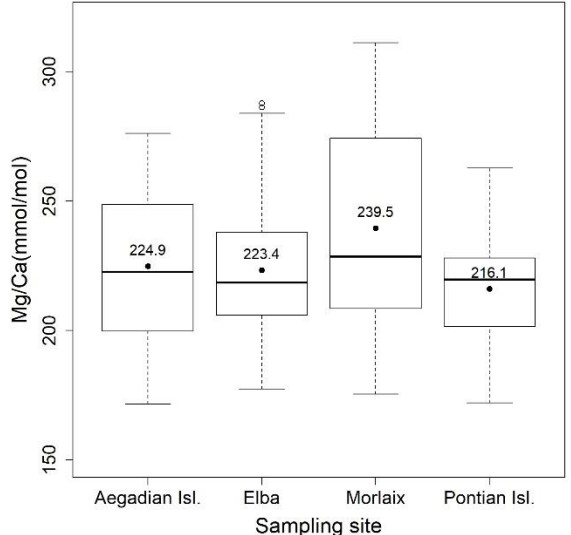

**Figure 4: Bar plot of the statistical tests performed to evaluate the differences of Mg/Ca in *L. corallioides* collected in different sampling sites. The horizontal black lines indicate the median values. The black filled circles and the numbers inside the plot indicate the mean values.**

Long cells were identified as spots with high Mg/Ca and positioning on light growth bands. Conversely, short cells were located in dark growth bands and corresponded to low Mg/Ca ratio.



The ANOVA test followed by the Tukey's test for multiple comparisons evidenced a significant variability of algal Mg/Ca among 3 sites in long cells (Table A2; Fig. 5). In the long cells of *L. corallioides* collected from Aegadian Isl. and Pontian

Isl., the Mg/Ca data showed quite similar distribution (Table A2; Fig. 5). The Mg/Ca of the alga from Pontian Isl. was the lowest (236.6 ± 14.0 mmol/mol) (Fig. 5). In Morlaix a higher Mg/Ca mean value (285.8 ± 18.0 mmol/mol) was registered, significantly different compared to Aegadian (247.5 ± 21.9 mmol/mol) and Pontian Isl. ($p \leq 0.001$, Table A2; Fig. 5). In short cells, the differences in Mg/Ca among samples were not statistically significant ($p > 0.05$; Table A1). The magnesium incorporation was slightly higher in Morlaix (207.6 ± 18.6 mmol/mol) and very similar between Aegadian and Pontian Isl.

samples (respectively 197.2 ± 12.8 mmol/mol and 196.8 ± 15.9 mmol/mol) (Fig. 5).

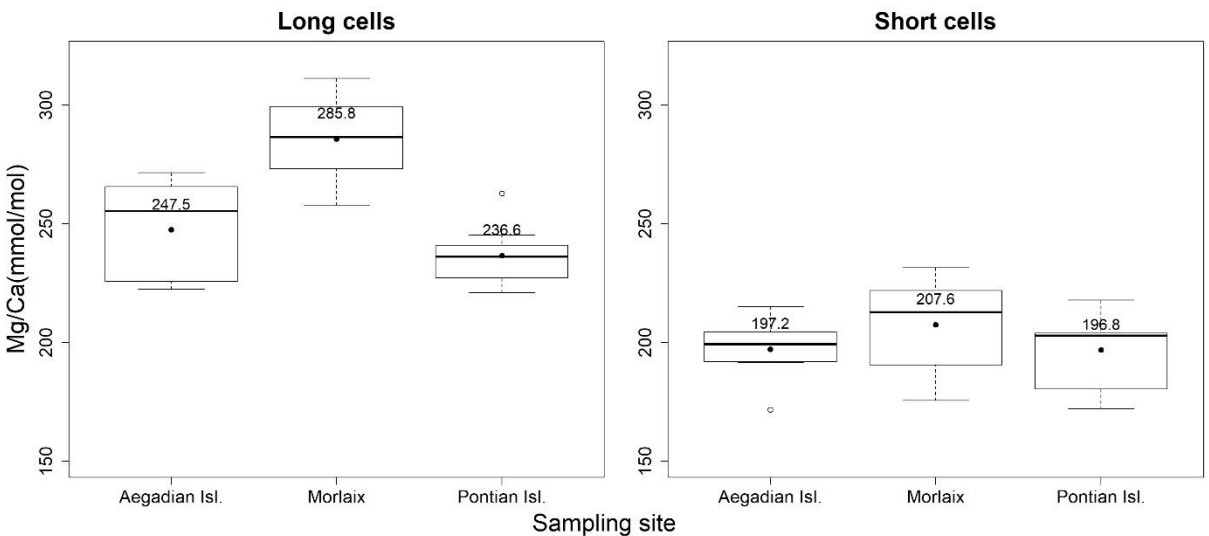

**Figure 5: Bar plot of the statistical tests performed to evaluate the differences of Mg/Ca in the long and short cells of *L. corallioides* collected in different sampling sites. The horizontal black lines indicate the median values. The black filled circles and the numbers inside the plot indicate the mean values.**

**3.3 B/Ca**

The B/Ca ratio in the sample collected from Morlaix showed a moderate positive correlation with all the examined temperature proxies (Mg/Ca, Li/Ca, Sr/Ca), with a more defined trend when plotted against Li/Ca ($r = 0.68$, $p < 0.001$) and slightly less defined against Mg/Ca ($r = 0.58$, $p < 0.01$) and Sr/Ca ($r = 0.57$, $p < 0.01$) (Fig. 6). On the contrary, the Spearman's analyses did not evidence significant correlations between B/Ca and the temperature signals in the algae collected elsewhere.

Overall, the B/Ca ratio in *L. corallioides* was 661.9 ± 138.9 μmol/mol, registering the minimum value in the long cells of the sample from Pontian Isl. (356.0 μmol/mol) and the maximum value in Elba (954.1 μmol/mol) (Fig. 7).

The Kruskal-Wallis coefficient evidenced a highly significant difference in the B/Ca value among sites ($p \ll 0.0001$), particularly in the *L. corallioides* from the Pontian Isl., which had the lowest boron incorporation (a mean B/Ca of 462.8 ±




49.2 μmol/mol) (Table A3; Fig. 7). The algae collected in Aegadian Isl. had still significantly lower B/Ca compared to those
collected in Elba and Morlaix; a mean of 610.8 ± 63.9 μmol/mol (Table A3; Fig. 7). The highest B/Ca mean value was registered in Elba (757.7 ± 75.5 μmol/mol), with medians comparable to Morlaix (726.9 ± 102.8 μmol/mol by mean) (Table A3; Fig. 7).

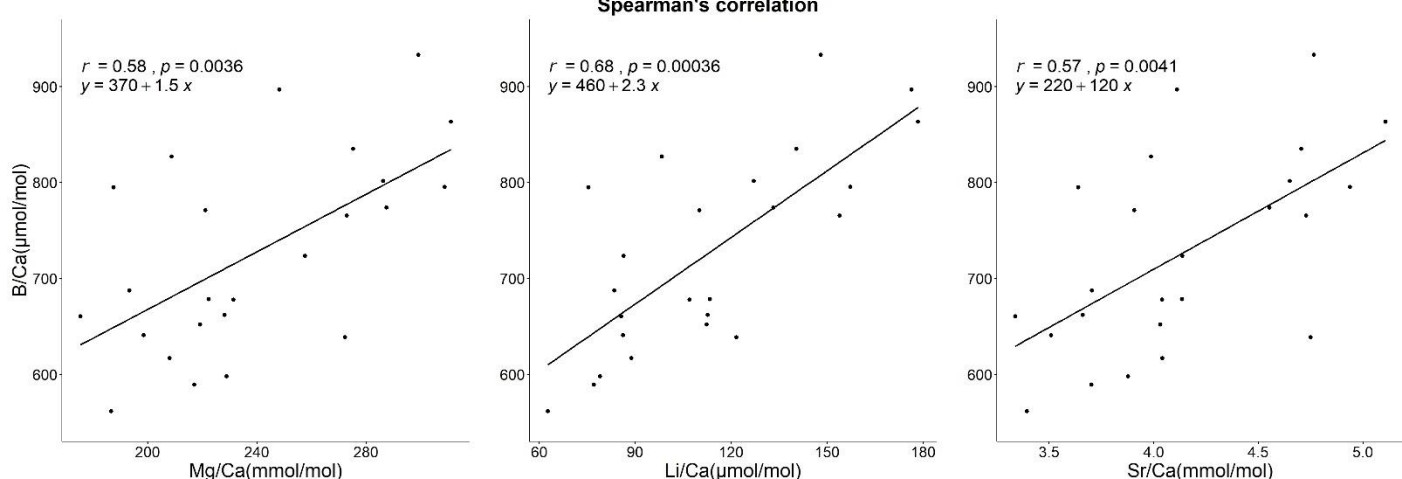

**Figure 6: Correlation plots of B/Ca with Mg/Ca, Li/Ca and Sr/Ca in *L. corallioides* collected in Morlaix bay. For each analyses the**
**Spearman's coefficient r, the p-value and the line equation are given.**

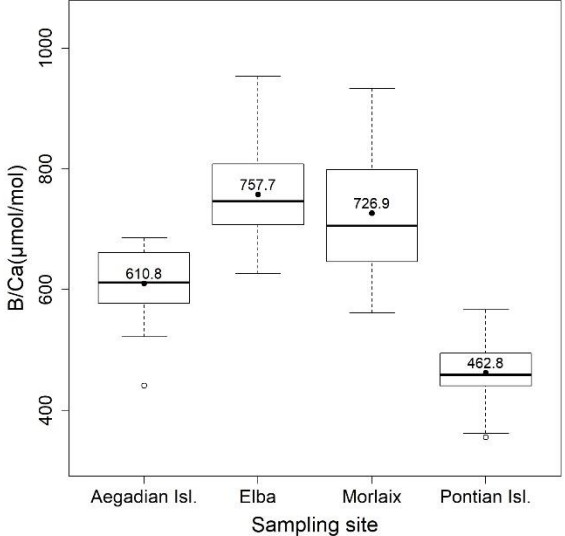

**Figure 7: Bar plot of the statistical tests performed to evaluate the differences of B/Ca in *L. corallioides* collected in different sampling sites. The horizontal black lines indicate the median values. The black filled circles and the numbers inside the plot indicate the mean values.**





The ANOVA test followed by the Tukey's test for multiple comparisons by site, for long (Table A4) and short cells (Table A5) separately, showed analogous trend (Fig. 8).

The algal cells from Pontian Isl. had the lowest mean B/Ca in both seasons (466.7 ± 58.9 µmol/mol in long cells and 460.8 ± 28.6 µmol/mol in short cells), being significantly different from both the samples from Morlaix and Aegadian Isl. (Table A4, A5; Fig. 8). Morlaix had the highest mean B/Ca in both long (792.3 ± 83.8 µmol/mol) and short cells (660.7 ± 69.3

µmol/mol) (Table A4, A5; Fig. 8). *L. corallioides* from Aegadian Isl. had intermediate B/Ca mean value in long cells (602.1 ± 93.5 µmol/mol), differing significantly from both the Morlaix and Pontian Isl. samples (Table A4; Fig. 8). In short cells, the sample from Aegadian Isl. slightly differed from the one in Morlaix, incorporating 617.7 ± 53.6 µmol/mol of boron by mean (Table A5; Fig. 8).

Interestingly, the long cells of all samples had higher median B/Ca values compared to short cells (Fig. 8), although only in

Morlaix, the differences between B/Ca measured in long and short cells were statistically significant ($\chi^2$=8.4899, p=0.0036).

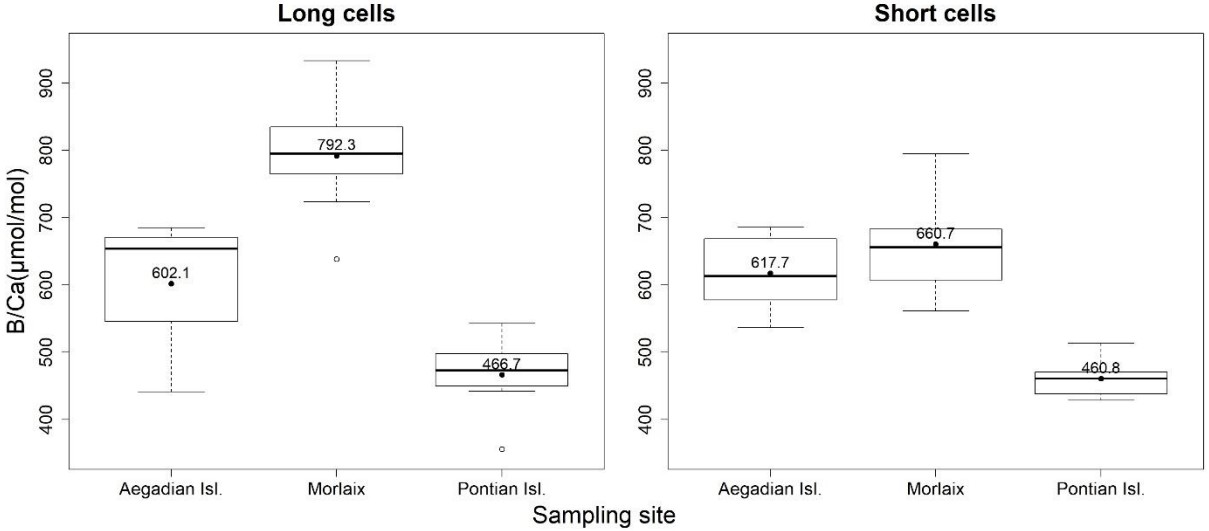

**Figure 8: Bar plot of the statistical tests performed to evaluate the differences of B/Ca in the long and short cells of *L. corallioides* collected in different sampling sites. The horizontal black lines indicate the median values. The black filled circles and the numbers inside the plot indicate the mean values.**

**3.4 Growth rates**

In the sample from Aegadian Isl., the LA-ICP-MS transect was 1.31 mm long and 10 years of growth have been detected by coupling microscopical imaging and Mg/Ca peaks, resulting in 0.13 mm/yr of growth rate. In the Elba sample the laser transect was 1,15 mm long, crossing 8 years of growth, with a resulting growth rate of 0.14 mm/yr. The Pontian Isl. sample had 1.08 mm of transect including 11 years of growth, hence a growth rate of 0.10 mm/yr. Finally, the transect from Morlaix

sample was 1.38 mm long, counting 11 years and resulting in 0.13 mm/yr of growth rate.



Growth rates did not show any linear relationship with Mg, Li and Sr/Ca, but they were positively correlated with the samples mean B/Ca values (Fig. 9).

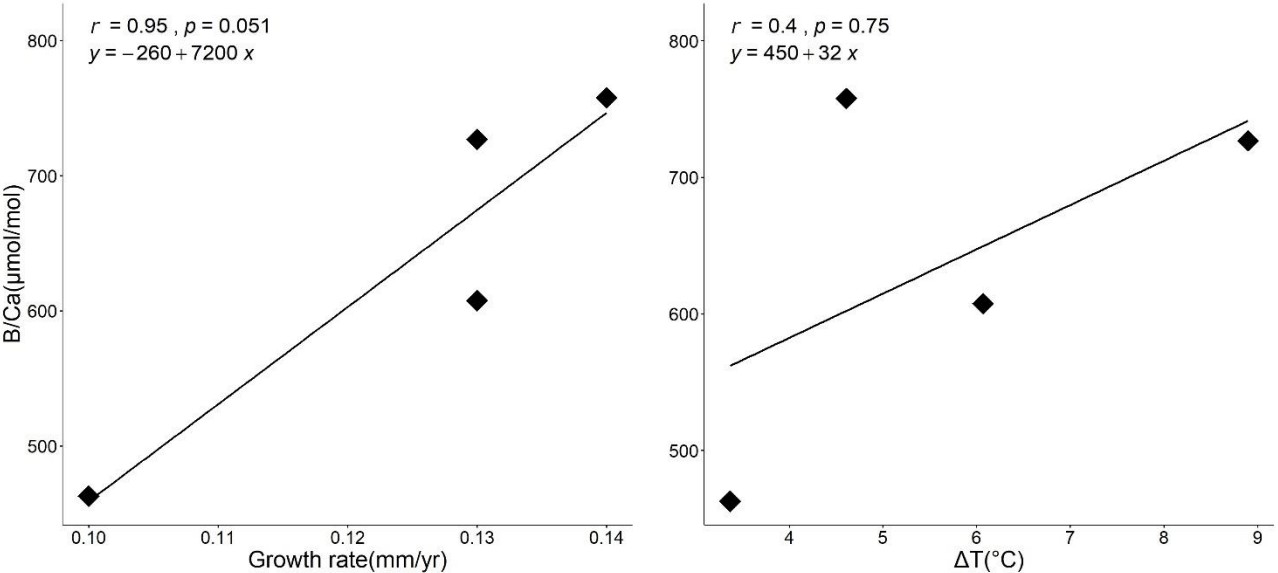

**Figure 9: Correlation plots of growth rates and seawater temperature variations (ΔT) with B/Ca in *L. corallioides* samples analysed in this study. Spearman's coefficient r, the p-value and the line equation are given.**

**4 Discussion**

In order to provide high-resolution geochemical data on long and short cells separately, we considered only the results from the spots where the positive/negative Mg/Ca peaks correspond respectively to the light/dark bands resulting from the image analyses. Hence, laser spots lying on light bands and with positive Mg/Ca peaks were identified as long cells; short cells were instead those included in dark bands and with negative Mg/Ca peaks. Thanks to this expedient, we avoided artefacts due to the Mg/Ca heterogeneity across the algal thallus (Fietzke et al., 2015; Nash and Adey, 2017a, b). In the sample from Elba the growth bands were not clearly visible, preventing the analyses of trace elements on long and short cells separately.

Trace elements concentrations recorded from the four *L. corallioides* branches analysed in this study were consistent with previously published values for other CA (Chave, 1954; Hemming and Hanson, 1992; Hetzinger et al., 2011; Darrenougue et al., 2014). Particularly, the range of Mg/Ca ratios resulted in this study extended from 172 to 311 mmol/mol, comparable to previous studies on rhodoliths of *Lithothamnion glaciale* Kjellman 1883 grown at 6-15 °C (148-326 mmol/mol) (Kamenos et al., 2008).

The B/Ca ratios in *L. corallioides* from our results range from 356 to 954 µmol/mol, higher than the range measured in *Neogoniolithon* sp. (352-670 µmol/mol) (Donald et al., 2017) and *C. compactum* (320-430 µmol/mol) (Anagnostou et al., 2019), both cultured with controlled $pCO_2$ and a pH ranging from 7.2 to 8.2. The paucity of B/Ca measurements from CA

 

and, most of all, the complete absence of these data on wild deep-water specimens make it difficult to compare our B/Ca data with the literature. This evidence takes stock of the significance of our results and emphasizes the importance of collecting more representative B/Ca data in CA. Nevertheless, Elba and Morlaix samples appeared to have extremely out of range B/Ca values (Fig. 7), compared to the range measured by Donald et al. (2017) and Anagnostou et al. (2019), suggesting the presence of diverse factors.

In general, temperature variations affect many physiological processes involved in the biomineralization and the rate of calcification influences the content of trace elements in carbonates, along with the preservation state of mineral structures (Lorens, 1981; Rimstidt et al., 1998; Gussone et al., 2005; Noireaux et al., 2015; Kaczmarek et al., 2016).

For the first time, we confirmed here the reliability of the temperature proxies Li/Ca and Sr/Ca on a deep-water Mediterranean CA. The results of the statistical analyses on Mg/Ca evidenced a strong relationship with the seawater temperatures extracted from ORAS5 (Table 2), as expected. *L. corallioides* from Aegadian Isl. had slightly higher Mg/Ca values, followed by Elba and Pontian Isl. (Fig. 4). This was consistent with local temperature values in the Mediterranean (Table 2), since Pontian Isl. registered the lowest mean value (14.89 °C) and the lowest temperature variation (ΔT) (3.37°C), while Aegadian Isl. showed the highest mean temperature (16.18 °C) and ΔT (6.07°C).

On the contrary, the sample from Morlaix, collected at 12 m depth, showed high Mg/Ca values in both long and short cells (Table A2; Fig. 5). The monthly mean temperatures had the highest variations during the year (ΔT in Table 2), due to the shallow depth (12 m). Temperature covaries with irradiance and both correlate to seasons, which influence primary production, respiration and calcification in *L. corallioides* (Payri, 2000; Martin et al., 2006) as well as other CA (Roberts et al., 2002). The high seasonality that characterized the sample from Morlaix, was responsible for the highest variation of Mg/Ca values and undoubtedly accounted for most of the differences with Mediterranean samples.

Li/Ca and Sr/Ca records were positively correlated with Mg/Ca in *L. corallioides* (Fig. 3), which, in turn, showed a strong relationship with seawater temperature. Therefore, Li/Ca and Sr/Ca could be regarded as temperature proxy in *L. corallioides*. The coupling of the Mg/Ca ratio with Li/Ca and Sr/Ca represents a multi-proxy approach that minimize the possible species effect and can be considered a useful tool to gather information about past temperature for paleoclimate reconstructions (Halfar et al., 2011; Caragnano et al., 2014; Williams et al., 2014; Fowell et al., 2016; Cuny-Guirriec et al., 2019).

B/Ca ratio in CA has been rarely measured and it is not clear how the environmental factors control its incorporation. The carbonate system primarily drives the changes in B incorporation (Hemming and Hanson, 1992; Yu and Elderfield, 2007). B/Ca, indeed, increases with $[CO_3^{2-}]$ (Yu and Elderfield, 2007) and [DIC] (Uchikawa et al., 2015), whereas there is no consensus on the effect of $[CO_3^{2-}]$ on Mg/Ca and Sr/Ca in benthic foraminifera (Rosenthal et al., 2006; Dueñas-Bohórquez et al., 2011). Nevertheless, the occurrence of high DIC concentrations in Elba (8.32 mol/m³) (Table 2) compared to most of the other sampling sites, caused the increase in B/Ca, but not in Mg/Ca and Sr/Ca. Our results showed indeed consistency between Sr/Ca and Mg/Ca data, without increased values in the sample from Elba. This evidence would exclude both Mg/Ca





and Sr/Ca dependence on DIC variations in the CA *L. corallioides*, contrarily to the results of Donald et al. (2017) in
*Neogoniolithon* sp. and Keul et al. (2017) in foraminifera.

Nevertheless, pH and DIC results (Table 2) did not explain our B/Ca results in the samples from Morlaix and Pontian Isl.
Significantly higher values of B/Ca should be expected in the sample from Pontian Isl., with the same pH and DIC as Elba
(8.13; 2.32 mol/m$^3$), as well as lower values in the sample from Morlaix, where pH and DIC were the lowest (respectively
8.06 and 2.17 mol/m$^3$).

The estimated growth rate of *L. corallioides* was $0.13 \pm 0.02$ mm/yr and it was supposed to decrease with increasing depth as
a direct consequence of a lower light availability (Halfar et al., 2011); indeed, the growth rate of the sample from Pontian Isl.
was the lowest (0.10 mm/yr). As already proved by previous studies on both synthetic and biogenic calcite, B incorporation
is likely affected by growth rate (Gabitov et al., 2014; Mavromatis et al., 2015; Noireaux et al. 2015; Uchikawa et al., 2015;
Kaczmarek et al., 2016). Indeed, in the cultured CA *Neogoniolithon* sp. the B/Ca increases with increasing growth rate
(Donald et al., 2017). In Pontian Isl., the slow growth rate probably contributed to the low B/Ca values. On the contrary, the
mean annual growth rate of the shallowest sample (Morlaix) was equal to the one in Aegadian Isl. (0.13 mm/yr). In Morlaix,
the alga probably significantly slowed down the growth in cold months, when the monthly mean seawater temperature was
the lowest of all the sampling sites (8.28 °C) (Table 2). Nevertheless, its growth rate likely speeded up in the warm season
due to the abundant light availability at shallow depth and the warming of seawater (Table 2), contributing to the
significantly higher B/Ca values in long cells (Fig. 8). Thus, the effect of the growth rate appeared evident from the
shallowest to the deepest samples (Fig. 9), because of the positive correlation between growth rates and mean B/Ca values.

Our results suggest that temperature is, in a way, related to B/Ca ratio in *L. corallioides*. Indeed, B/Ca ratio decreased with
$\Delta T$ across depth (Fig. 7), with the exception of the Elba sample. In Morlaix, B/Ca showed higher values in comparison with
Aegadian Isl. and Pontian Isl. ($726.9 \pm 102.8$ μmol/mol), and a positive correlation with temperature proxies (Mg/Ca, Li/Ca
and Sr/Ca; Fig. 6). A positive correlation between B/Ca and Mg/Ca was already observed in planktonic foraminifera (Wara
et al., 2003; Yu et al., 2007).

In the sample from the Pontian Isl., the seasonal $\Delta T$ (3.37°C), Mg/Ca ($216.1 \pm 21.9$ mmol/mol) and B/Ca ($462.8 \pm 49.2$
μmol/mol) values were the lowest among sites. In particular, B/Ca was significantly low (Fig. 7), differing more from the
other samples than the results on Mg/Ca (Fig. 4), suggesting that in this sample the B incorporation could be influenced by
other factors. In general, the poor correlation with temperature (Fig. 9), especially in deep-water samples, excludes the
suitability of B/Ca as a temperature proxy and suggests a closer relationship with growth rate rather than temperature.

Knowing the biogeochemistry and the variation of the environmental parameters of seawater is crucial for a more
comprehensive picture of the reliability of geochemical proxies, like the ones we investigated in this paper (Mg, Li, Sr/Ca
and B/Ca). Boron incorporation in marine carbonates is still debated, rising questions about the boron isotopic fractionation,
the mechanisms of boron incorporation into marine carbonates, the so-called "vital effects" (i.e. the metabolic activities that
can bias the isotopic signal), and the seawater isotopic composition. Further studies on *L. corallioides* and other CA should





be carried out to clarify the environmental factors influencing the substitution of boron within the calcite lattice of these organisms, prior to adopting boron-based proxies for paleoclimate reconstructions.

## 5 Conclusion

This paper presented the first measures on trace elements (Mg, Sr, Li and B) from the CA *L. corallioides* collected across the Mediterranean Sea and in the Atlantic Ocean, at different oceanographic settings and depths.

LA-ICP-MS records of Mg/Ca, Sr/Ca and Li/Ca have shown a similar trend, primarily controlled by seawater temperatures in the algal habitat. Indeed, higher Mg/Ca values were registered in Morlaix and lower Mg/Ca in Pontian Isl., which had respectively the highest and lowest ΔT.

In order to evaluate the control exerted by temperature on B incorporation, we also tested the correlation between B/Ca with Mg/Ca, Li/Ca and Sr/Ca. This led us to provide the first B/Ca data on wild grown deep-water CA. The correlation between B/Ca and Mg/Ca in *L. corallioides* was statistically significant only in the shallow waters of Morlaix, where seasonality, hence the seasonal temperature variations, during the algal growth was the strongest among the samples studied (8.90 °C). Accordingly, B incorporation differences between long and short cells of *L. corallioides* strongly depend on the magnitude

of temperature fluctuations, being statistically significant just in Morlaix. The extracted carbon data did not explain the low B concentration in the deepest sample, Pontian Isl. (462.8 ± 49.2 μmol/mol), where pH and DIC were high compared to the other sampling sites (8.12; 2.32 mol/m$^3$). We also found high B/Ca values in the sample from Morlaix (726.9 ± 102.8 μmol/mol), at shallow depth, where pH and DIC were the lowest (8.09; 2.17 mol/m$^3$). The estimation of growth rate, that is low in the deepest sample (Pontian Isl., 0.11 mm/yr) and gets higher in the Morlaix samples (0.13 mm/yr), led us to conclude

that B/Ca relates to growth rate rather than seawater temperature. In deep samples, B incorporation decreases with ΔT and depth, as growth rates did. Indeed, no statistically significant correlation can be traced with other temperature proxies.

B incorporation is therefore subject to the specific algal growth patterns and rates, whose knowledge is essential in order to assess the reliability of B/Ca in tracing seawater carbon variations.

## Appendix A

**Table A1: (a) Statistically non-significant results of tests performed to evaluate (a) the differences of Mg/Ca in *L. corallioides* and (b) the differences of Mg/Ca in the short cells of *L. corallioides* collected in different sampling sites. Test significance at α = 0.05.**

| (a) | Kruskal-Wallis test (Mg/Ca) | | |
|---|---|---|---|
| | Df | $\chi^2$ | P |
| SITE | 3 | 3.799 | 0.284 |

| (b) | One-way ANOVA test (Mg/Ca) | | |
|---|---|---|---|





| Short cells | | | | | |
|---|---|---|---|---|---|
| | Df | Sum sq. | Mean sq. | F value | Pr(>F) |
| SITE | 2 | 788.1 | 394.0 | 1.4647 | 0.2496 |
| Residuals | 26 | 6994.5 | 269.0 | | |
| Shapiro-Wilk normality test | | | P=0.6442 | | |
| Bartlett's K-squared | | | P=0.5856 | | |

**Table A2: Results of statistical tests performed to evaluate the differences of Mg/Ca in the long cells of *L. corallioides* collected in different sampling sites. Statistically significant p-values are given in bold. ANOVA test significance at α = 0.05; Tukey's test significant at p ≤ α.**

| One-way ANOVA test (Mg/Ca) | | | | | |
|---|---|---|---|---|---|
| Long cells | | | | | |
| | Df | Sum sq. | Mean sq. | F value | Pr(>F) |
| SITE | 2 | 10897.7 | 5448.9 | 16.413 | **0.0001** |
| Residuals | 20 | 6639.8 | 332.0 | | |
| Shapiro-Wilk normality test | | | P=0.1440 | | |
| Bartlett's K-squared | | | P=0.5826 | | |
| **Tukey's test** | | | | | |
| Multiple comparisons of means | | | | | |
| SITE | Mean difference | 95% confidence interval | | P. adjusted | |
| | SITE | lower bound | upper bound | | |
| Morlaix-Aegadian Isl. | 38.32918 | 15.09816 | 61.56019 | **0.00130** | |
| Pontian Isl.-Aegadian Isl. | -10.84361 | -35.48382 | 13.79661 | 0.51716 | |
| Pontian Isl.-Morlaix | -49.17278 | -72.40380 | -25.94177 | **0.00009** | |

**Table A3: Results of statistical tests performed to evaluate the differences of B/Ca in *L. corallioides* collected in different sampling sites. Statistically significant p-values are given in bold. Kruskal-Wallis test significance at α = 0.05; Dunn's test significant at p ≤ α/2.**

| Kruskal-Wallis test (B/Ca) | | | |
|---|---|---|---|
| | Df | $\chi^2$ | P |
| SITE | 3 | 79.816 | **<2.2e-16** |
| **Dunn's test** | | | |
| Comparisons by SITE (Bonferroni) | | | |
| Z | Aegadian Isl. | Elba | Morlaix |





| P. adjusted | | | |
|---|---|---|---|
| Elba | -4.64580 | | |
| | **0.00000** | | |
| Morlaix | -3.07755 | 1.17249 | |
| | **0.00630** | 0.72300 | |
| Pontian Isl. | 2.80564 | 8.38673 | 6.15663 |
| | **0.01510** | **0.00000** | **0.00000** |

**Table A4: Results of statistical tests performed to evaluate the differences of B/Ca in the long cells of *L. corallioides* collected in different sampling sites. Statistically significant p-values are given in bold. ANOVA test significance at α = 0.05; Tukey's test significant at p ≤ α.**

| One-way ANOVA test (B/Ca) | | | | | |
|---|---|---|---|---|---|
| Long cells | | | | | |
| | Df | Sum sq. | Mean sq. | F value | Pr(>F) |
| SITE | 2 | 428364 | 214182 | 33.066 | **0.0000** |
| Residuals | 20 | 129546 | 6477 | | |
| Shapiro-Wilk normality test | | | P=0.5527 | | |
| Bartlett's K-squared | | | P=0.5470 | | |
| **Tukey's test** | | | | | |
| Multiple comparisons of means | | | | | |
| SITE | Mean difference | 95% confidence interval | | P. adjusted | |
| | SITE | lower bound | upper bound | | |
| Morlaix-Aegadian Isl. | 190.11730 | 87.50374 | 292.73094 | **0.00040** | |
| Pontian Isl.-Aegadian Isl. | -135.42490 | -244.26303 | -26.58672 | **0.01342** | |
| Pontian Isl.-Morlaix | -325.54220 | -428.15581 | -222.92862 | **0.00000** | |


**Table A5: Results of statistical tests performed to evaluate the differences of B/Ca in the short cells of *L. corallioides* collected in different sampling sites. Statistically significant p-values are given in bold. ANOVA test significance at α = 0.05; Tukey's test significant at p ≤ α.**

| One-way ANOVA test (B/Ca) | | | | | |
|---|---|---|---|---|---|
| Short cells | | | | | |
| | Df | Sum sq. | Mean sq. | F value | Pr(>F) |
| SITE | 2 | 216232 | 108116 | 35.360 | **0.0000** |
| Residuals | 26 | 79497 | 3058 | | |





| Shapiro-Wilk normality test | | | P=0.1699 | |
|---|---|---|---|---|
| Bartlett's K-squared | | | P=0.0576 | |
| **Tukey's test** | | | | |
| Multiple comparisons of means | | | | |
| SITE | Mean difference | 95% confidence interval | | P. adjusted |
| | SITE | lower bound | upper bound | |
| Morlaix-Aegadian Isl. | 43.09640 | -19.61932 | 105.81212 | 0.22146 |
| Pontian Isl.-Aegadian Isl. | -156.90170 | -223.66771 | -90.13574 | **0.00001** |
| Pontian Isl.-Morlaix | -199.99810 | -260.58727 | -139.40898 | **0.00000** |

**Data availability**

Data resulting from this study are available from the authors upon request to the corresponding author.

**Author contributions**

DB, VB, and GP conceptualized the research question and study design. AL, DB, and VB conducted the experimental work; AM and GP the environmental data extraction. GP performed the data analysis and prepared the draft of the paper. All authors contributed to the editing and reviewing of the paper.

**Competing interests**

The authors declare that they have no conflict of interest.

**Acknowledgements**

This paper is a contribution to the Project MIUR-Dipartimenti di Eccellenza 2018-2022 DISAT-UNIMIB.
The Pontian Isl. sample has been collected in the framework of "Convenzione MATTM-CNR per i Programmi di
Monitoraggio per la Direttiva sulla Strategia Marina (MSFD, Art. 11, Dir. 2008/56/CE)". Captain, crew and scientific staff of RV Minerva Uno cruise STRATEGIA MARINA LIGURE-TIRRENO are acknowledged for their efficient and skilful cooperation at sea.
Financial support for GP was provided by Italian MIUR as a PhD fellowship.
Environmental data were provided by E.U. Copernicus Marine Service Information.





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
