# Peer review of "Growth rate rather than temperature affects the B/Ca ratio in the calcareous red alga *Lithothamnion coralliooides"

_Biogeosciences, 2021_

## Author Comment (AC2)

[Figure]

**Figure S1: Longitudinal section through the *L. corallioides* branch sampled in (a) Aegadian Isl. (scale bar = 100 µm), (b) Elba (scale bar = 350 µm) and (c) Pontian Isl. (scale bar = 200 µm), showing the LA-ICP-MS transects targeting each growth bands.**

---

## Author Response (AR1)

Dear Referee #1,

Thank you for your accurate review and for the time you spent in carefully reading our work. Our paper has benefited a lot from your suggestions. We are also glad for your appreciation on the scope of our research and that you recognize the uniqueness of our data.

Hereafter I will reply to each of your comments.

- **Comment:** *The authors say in the abstract "We produced the first data on temperature proxies (Mg, Li and Sr/Ca)" but it would be really nice to see more made of these important data. If these results are to be used to establish temperature dependency (or not) on B/Ca, the temperature dependency of Mg Li and Sr must first be convincingly made first. Many papers are cited here that show the correlation between these variables on seasonal cycles, but it has not been established that temperature is the main driver of the incorporation over seasonal growth rate. The use of Li/Ca in particular as a "temperature proxy" requires more consideration. The Darrenougue paper cited does indeed shows that Li/Ca is high in the warm summer months, but it is not clear that this is driven by temperature. Li has a very low partition coefficient, so as growth rates increase during summer kinetics and growth entrapment models would suggest that Li/Ca go up. This growth rate dependency may mask the Li/Ca temperature effect which some have argued is a negative slope in carbonates and should be ratioed to Mg for reliable temperatures (Anagnostou et al., 2019; Stewart et al., 2020 EPSL; Marriott et al., 2004 EPSL and Chem Geol.). It would be interesting to see where these Li/Mg data fall on calibration lines of Anagnostou et al., 2019 and Stewart et al., 2020 (EPSL). Do they agree with the linear fit of Anagnostou for cultured CCA or the speculative exponential relationship found across all high-Mg calcites suggested in Stewart et al., 2020? The Li/Mg results here (I am estimating as raw Li/Ca ratios were not provided in a table) seem perhaps a little high compared to Anagnostou culture data, but this could be due to analytical offsets that can occur during laser ablation analysis compared to solution chemistry.*

   **Response:** We agree with you that it would be extremely interesting to quantify the temperature and growth rate dependency on Mg, Li and Sr. We also agree that the application of the Li/Ca proxy is still controversial, whereas Mg/Ca has a widely recognized reliability as temperature proxy in coralline algae. Nevertheless, Li/Ca has been proven to be a reliable seawater temperature proxy in coralline algae (Caragnano et al., 2014; 2017, Anagnostou et al., 2019), and a Mg/Li calibration did not improve the Mg/Ca or Li/Ca temperature relationship in previous field studies (Caragnano et al., 2014; 2017). We have plotted our Mg/Li data on the calibration line of Anagnostou et al. (2019), as you suggested (Figure 6). As you can see, our data fall in the range of values found by those authors. Nevertheless, our Mg/Li results did not show significant correlation with temperature. This has been evidenced also by the lack of seasonal oscillations in Mg/Li values found in Morlaix (Figure 7). Therefore, Mg/Li is not considered a significant temperature proxy for the purpose of this work. These results have been added to the revised paper. In our paper, Li/Ca data (as well as Sr/Ca) have a significant positive correlation with Mg/Ca, which is considered mainly controlled by temperature. Therefore, we confirm other studies which suggest a dominant control of temperature also in the Li incorporation. Anagnostou et al. (2019), which you cited, performed geochemical analyses on the cultured coralline algal species *Clathromorphum compactum* and found an inverse relationship between Li/Ca and seawater temperature. We recognize the importance of culture studies to understand the geochemical equilibrium between these calcifying organisms and seawater chemistry. Nevertheless, data collected in the field on wild-grown specimens, do not allow us to perform such high-resolution calibrations, that were beyond our research purpose. This point will be further discussed in the next responses.

- *Comment: Do the short or long cells measured offer a better temperature reconstruction? It would be great to see the authors bring this into the discussion and would add to the interpretation of these elemental ratios as being temperature dependent thus permitting their use to unpack competing effects on B/Ca.*

  **Response**: The separation of short and long cells definitely allowed to showcase the correlation of skeletal pattern and Mg incorporation. We agree that there are multiple factors influencing the element/Ca in coralline algae, as well as in other calcifying organisms. Nevertheless, to measure the influence of each single factor, laboratory experiments would be needed. In our paper, the distinction between the geochemical signal from short and long cells allowed us to evaluate the banding variations in the element content. This was useful to investigate, for the first time, the B/Ca variability across the algal thallus. Other considerations are object of ongoing investigation and beyond the aim of this contribution.

- *Comment: Of particular concern, the main finding that growth rate is driving B/Ca is based on a correlation of just 4 data points (figure 9), three of which have very similar growth rates. The lack of variety of environmental conditions and growth rates means that, despite the presentation of p values, the significance of this relationship is highly questionable. I cannot see that the authors are in a position to establish the outright controls on B/Ca with the data collected. The 4 samples presented here, 3 of which are in the Med, simply do not cover a wide enough array of hydrographic conditions to pick these effects apart.*

  **Response**: We believe that the strength of our paper relies on the fact that it is the first work comparing the same coralline algal species across a natural gradient, across different depths and even different Basins. This implies both strengths and limits, since it allowed us to observe the response of a single widely distributed species in natural environmental conditions, but with a limited set of variations compared to culture experiments in the laboratory-controlled conditions. Moreover, we would like to note that one of the main differences in working with wild-grown specimens is the impossibility to control and check for each environmental factor influencing the calcification during growth. As you highlighted above, there are multiple factors influencing the geochemistry of these organisms and the best that we can do is to collect as much data as possible to take the big picture, then identifying the main contributing factors to the variations we observe. No coralline algae live in any possible environment, and actually our "4 points" derive from several observations and several samples collected in the natural environment. Please note that the Mediterranean is not an experimental tank in controlled conditions, because it shows a wide range of possible combination of environmental controls. In other words, the pattern shown in Fig 9 has an intrinsically strong significance for *L. corallioides*.

- *Comment: Given the uncertainty on driving factors of all of these elements, not just B, I would recommend the authors take a more explorative approach to their results rather than categorically stating which are temperature dependent. It would be useful to see all elemental data plotted as a time series to see if there are trends along the growth axis and the cyclical nature of each elemental ratio with season. This could be compared to the CA image to show the matching cycles to light and dark banding. It would be great to see these annual banded timeseries matched to gridded SST records for each site (e.g. ERSST https://www.ncdc.noaa.gov/data-access/marineocean-data/extended-reconstructed-seasurface-temperature-ersst-v5 or HADISST https://www.metoffice.gov.uk/hadobs/hadisst/data/download.html ). This would really help to convince the reader about the temperature control on each element.*
  *In short, I would very much like to see these important data published, but with a more*

*measured approach to the discussion.*

**Response**: We agree on the complexity of interpreting the driving factors of element/Ca and the text has been modified for less straight forward conclusions. We also agree that the paper could benefit from adding a plot showing the time series of all elemental data, highlighting the elemental cyclicity, as you suggested. We have therefore provided the new plot and we thank you for your suggestion (Figure 12).

- **Comment:** *Line 16: "from shallow to deep waters" seems misleading. None of these are truly deep waters. Perhaps "from across the photic zone depths" would be better.*

  **Response**: The text has been modified.

- **Comment:** *Line 22: "This evidence suggests"*

  **Response**: The text has been modified.

- **Comment:** *Line 32: $pK_B$ is pressure, temperature, and salinity dependent therefore the value of 8.6 is only applicable to surface waters (c.f. the value in deep waters is 8.8). Be clear that this is not a constant and that this refers to "typical surface seawater conditions"*

  **Response**: This has been specified in the text as suggested.

- **Comment:** *Line 35: The Klochko value for the fractionation factor is more commonly given in the 1.0272 format.*

  **Response**: The text has been modified.

- **Comment:** *Line 35: "Seawater isotopic composition $\delta11B_{sw}$ is 39.61‰ (Foster et al., 2010) and varies with the isotopic composition of B(OH)3 and B(OH)4-, both enriched in $\delta11B$ with increasing pH (Dickson, 1990)." This sentence is poorly worded. Seawater $\delta11B$ does not vary with the isotopic composition of boric acid and borate. Rather the $\delta11B$ of seawater is fixed and the 11/10 ratios of boric acid and borate have to change to maintain the fractionation factor as their abundances change with changing seawater pH.*

  **Response**: The text could be misunderstanding and has been modified for clarity as you suggested.

- **Comment:** *Line 50: It is worth stating that these B/Ca vs CO32- relationships are only empirically derived observations.*

  **Response**: This has been specified as suggested.

- **Comment:** *Line 52: Paragraph starting "Warming and acidification…" should be the opening of the introduction leading to a statement about proxies for the carbonate system are therefore much needed. The discussion of $\delta11B$ and B/Ca (currently starting at line 25) will then follow better.*

  **Response**: This change has been made.

- **Comment:** *Line 55: Unclear what is meant by "longevity by indeterminate growth,". As outlined in the next sentence many specimens have annual growth bands therefore we can determine their growth axis and age.*

  **Response**: The "indeterminate growth" is a botanical concept that has been widely referred to coralline algae (Adey, 1965; Frantz et al., 2005; Halfar et al., 2008), and it is held responsible of their longevity. It contrasts with the growth pattern of bivalves, for example, which are influenced by an ontogenetic growth trend which slows down with age following an asymptotic curve. This concept is very important for climate reconstructions since an ontogenetic trend would compromise the resolution of the geochemical signals in the later stages of growth. Coralline algae, instead, do not slow growth over time and the element incorporation is not governed by age. We understand that this concept should be better explained for non-specialists and we added a more detailed explanation in the text along with the citation.

- **Comment:** *Line 78: "culture experiment"*

  **Response**: The text has been changed.

- **Comment:** *Line 80: Suggested rewording: "The factors which influence the B incorporation in CA are therefore still debated."*

  **Response**: The text has been changed.

- **Comment:** *Line 88: I think Hetzinger et al., 2011 should be Hetzinger et al., 2009. The 2011 paper is about Ba rather than Mg Li or Sr*

  **Response**: Actually Hetzinger et al., 2011 shows Mg and Sr data besides Ba, while Hetzinger et al., 2009 is only about Mg/Ca.

- **Comment:** *Table 1 and 2 could be combined, that way the reader can see how the collection depth compares to the Temperature data. Also strange to have table 2 A and B – all of these columns could be combined into one table. Please consider if all columns are necessary. For example, Lat and Long could be moved into the caption of Fig1. If you have Temp max and min, do you need to spell out the range? Do you even need the range when you have the st. dev? Is standard dev not a better measure of the variability at each site than the range and demonstrates the point about the Atlantic being more variable on its own? It is also strange to present this info in a table and then take up words spelling it all out again in the main text (section 3.1 can be very much shortened).*

  **Response**: Table 1 and 2 have been combined as you suggested. Concerning the temperature range, we believe it is important to explicit these data since they have a different meaning from standard deviation. ΔT has indeed been measured as the difference between maximum and minimum temperature values extracted in the sampling site during the considered time interval. These data better explain the different temperature variations experienced by the alga during its growth and are directly related to the sampling depth. We agree that section 3.1 could be shortened just citing the data in the table and changes have been made to the text accordingly.

- **Comment:** *It would be nice to see all trace metal data tabulated somewhere too.*

**Response**: A new table has been added summarizing the elemental data resulting from the work (Table 2). Moreover, raw data have been submitted to the PANGAEA repository.

- **Comment:** *Table 2 caption. It is not clear what is meant by data "elaboration". Please consider wording here.*

**Response**: The text has been modified for clarity.

- **Comment:** *Line: 118: "Particularly,…" should read "In particular,…"*

**Response**: The change has been made.

- **Comment:** *Line 121: Suggest rewording "Once its inclusion under the genus Phymatolithon was excluded, the Morlaix sample…"*

**Response**: The text has been modified as you suggested.

- **Comment:** *Line 133: Suggest "Element/Ca ratios were calculated for these isotopes,"*

**Response**: The text has been modified.

- **Comment:** *Line 133: What about these elemental ratios was "in agreement with Yu et al., (2005) and Darrenougue et al., (2014)"? Presumably Yu and Darrenougue didn't measure these same samples, but did they measure the same standards? Was it the methods that are similar? It is not clear what is mean here.*

**Response**: Those authors measured the same isotopes in different samples. We have removed this sentence which could be misunderstood and does not add significant information to the methods.

- **Comment:** Fig 2: Please label long and short cells on panel b

**Response:** We have added the labels.

- **Comment:** *Line 140: this method section requires more information. What is the measurement accuracy and precision? NIST 612 was used as an "external standard" (does this mean a bracketing standard?), but were any other reference materials used to demonstrate measurement accuracy (e.g. JCp-1 coral powder pressed as a pellet)? Bracketing with NIST glass for carbonate laser ablation is far from ideal in terms of matrix matching, but others have demonstrated that it can give reasonable results (Fietzke et al., 2010).*

**Response:** We did not analyse other standards for quality control. Following your suggestions, we added the references of Fietzke et al. (2010) and also Jochum et al. (2012), which demonstrated the reliability of NIST glass standards.

- **Comment:** *Line 150: Suggest: "Carbonate system parameters for each site have also been estimated"*

**Response:** The text has been changed.

- **Comment:** *Line 160: suggest: "Growth rates were estimated under light microscope by measuring…"*

**Response:** The text has been changed.

- **Comment:** *Section 2.5. Suggest simpler subtitle "Statistical analysis"*

  **Response:** The subtitle has been modified.

- **Comment:** *Line 178: Lower amplitude seasonal temperature change. The word "excursion" implies an aberration from the norm, this doesn't fit when describing an annual cycle.*

  **Response:** The text has been corrected.

- **Comment:** *Line 179: "site" rather than "one"*

  **Response:** The text has been changed.

- **Comment:** *Line 189: The pH gradient described here doesn't really exist as described. pH estimates at the Mediterranean sites are all similarly high ~8.13 and less variable than the Atlantic site (~8.06). Similarly for DIC, as this is largely dictating the pH*

  **Response:** The results have been reported in the text following your suggestion.

- **Comment:** *Fig 3: p values of 2.2e-6 could just be written as p<0.01 as the extra decimal places after the 99% offer little additional information to the reader. Also no need to repeat these R and p values in the main text if they are in the figure. Title of "spearman correlation" should be removed as figure titles are redundant with a caption and this info is already in the caption. It this all data or one CA sample? It would be interesting to see how points vary by location. Perhaps colour code the points or use symbols for each site (this could be applied throughout to all figures, including the map, for consistency). Change Y and X of regression lines to Mg/Ca and Li/Ca*

  **Response:** All your suggestions on Figure 3 have been followed and the plot has been changed accordingly.

- **Comment:** *Section 3.2 more repetition of mean values that are shown in Figure 4. Consider making the Results section more concise throughout using the information presented in the figures and tables rather than repeating.*

  **Response:** The repetitions have been deleted throughout the text of the Results sections, improving readability. Thank you for your suggestion.

- **Comment:** *Line 216: "areas" rather than "spots" and "within light growth bands" rather than "positioning on"*

  **Response:** The text has been changed.

- **Comment:** *Box plot figures would make more sense if the sites were ordered Aeg, Elba, Pontian, and then Morlaix so that the Med sites are next to each other and the Atlantic site. Given the similarity of the Med sites hydrographic data effectively two environments are being compared: the Med and the Atlantic.*

**Response:** Box plot figures have been modified as you suggested.

- **Comment:** *Line 231: see mainpoint above. It is important to establish which direction Mg, Li, and Sr are expected to change as temperature changes. It remains unclear to what extent these elements themselves are driven by growth rate and to what extent they are driven by temperature. A full discussion of their partition coefficients to establish their respective response to growth rate is required before temperature effects can be established and used to unpack the effects on B/Ca.*

**Response:** As already explained in the response of your main point above, the information collected directly from a natural environment arises interpretation challenges. Our approach was to test the proxies in natural conditions where we cannot monitor the parameters influencing the algal growth as we could do in a controlled experimental setting. Therefore, we inferred the dependence on growth rate and temperature using the data we possessed without the possibility to perform calibrations and the discussion you were referring to. We decided to take reasonable approximations. Mg/Ca is considered a confident temperature proxy in coralline algae and we have no reason to doubt its reliability, since our results also show cyclic variations through time which are evidently temperature related (Figure 12). Considerations on Li and Sr/Ca temperature dependence had followed accordingly. We do not have the evidence to assume a major control of growth rates rather than temperature on Mg, Li and Sr incorporation.

- **Comment:** *Line 237: "shows" rather than "evidenced"*

**Response:** The text has been changed.

- **Comment:** *Line 251: Showed an analogous trend to what? Please be clear here*

**Response:** The text has been modified in "showed lower values in the Mediterranean sites and higher values in the Atlantic site".

- **Comment:** *Section 3.4 these extension rate results should be included in the combined table 1 and 2. It is noteworthy that the growth rate at Pontian was lower despite its similarities in other environmental conditions with the other Med sites.*

**Response:** The growth rates have been added in the new combined Table (Table 1).

- **Comment:** *Line 277 This begs the question how often did light and dark band match Mg/Ca patterns. A figure would be useful to show this matching to convince the reader that there is indeed a Mg/Ca pattern that fits the colour banding.*

**Response:** The plot showing the cyclical variations of Mg/Ca and other elements across the algal thallus has been added as mentioned before (Figure 12).

- **Comment:** *Figure 9: As with all figures, please identify the sites by colour or data point.*

**Response:** Figures have been modified according to your suggestions.

- **Comment:** *Line 280: consider wording here. Mg/Ca ratios do not go "negative". Suggest high and low*

**Response:** Positive/negative peaks have been changed to high/low as suggested.

- **Comment:** *Line 287 to 295: It is not surprising that the ranges in B/Ca measured here using a laser (capable of picking up more heterogeneity within samples) are wider than the bulk carbonate analyses measure by solution ICPMS performed by Anagnostou and Donald. Sure, it might be environmental, but the analytical difference must be acknowledged.*

  **Response:** Thank you for your comment. The different analytical methods have been pointed out in the text while comparing the literature data.

- **Comment:** *Line 299: I think this statement is rather unsupported. It has been established that Mg data covary with Li and Sr, not that Li and Sr are temperature dependent. This could be growth rate particularly given the statement on line 307 about temperature covarying with irradiance.. Also, again, the use of "deep water" is misplaced.*

  **Response:** As you pointed, we provided evidence that Mg data covary with Li and Sr. Nevertheless, since the major factor controlling Mg/Ca in coralline algae is temperature, this covariance suggests that temperature is also a determinant factor in controlling Li and Sr/Ca. Definitely, multiple factors contribute at the same time in the elements incorporation, including growth rates and light availability. In natural environments as in our study, we could not control all the factors influencing the algal growth and we interpreted the data we obtained without a constant monitoring. Moreover, the new plot showing the cyclical variations of elements/Ca through time, as you suggested to add, clearly shows a correlation between temperature and Mg, Li and Sr/Ca, contrary to B/Ca (Figure 12).

- **Comment:** *Line 319: This is also little consensus on the B/Ca effect in biogenic carbonates. New coral culture data by Gagnon et al., 2021 ESPL (see supplement) show that B/Ca increases with carbonate ion when DIC is held constant, and decreases with carbonate ion when pH is constant. This paper also shows a strong negative DIC effect on B/Ca (unlike the inorganic experiments of Uchikawa) however this may be strongly related to scleractinian biocalcification mechanisms and the rate of replenishment of the calcifying fluid with seawater.*

  **Response:** Thank you for the reference, which is very interesting. It will be cited in the text along with the negative correlation between [DIC] and B/Ca found by Donald et al. (2017) in a cultured coralline alga. This could contribute to the high B/Ca value found in Morlaix compared to the Mediterranean sites.

- **Comment:** *Line 321: check the DIC value. I think this should read 2.32 rather than 8.23. Also this value extremely similar to all of the other Med sites and is therefore unlikely to be driving B/Ca. Indeed the lowest DIC site (Morlaix) also has a high B/Ca (>700). This is acknowledged later on, but the statement about DIC causing the high B/Ca at Elba is completely unsupported by the results at the other sites.*

  **Response:** The typing error has been corrected and the discussion has been changed to better support the results, as mentioned in the previous response.

- **Comment:** *Line 333: "suggested" rather than "proved"*

  **Response:** The text has been changed.

- **Comment:** *Line 343: Similar to my point about the questionable significance of a correlation between just 4 data points, the relationship between B/Ca and ΔT could be considered absent or strongly positively covarying if Elba is considered anomalous. There are just not enough data to say either way. It is also not entirely clear why the magnitude of the seasonal cycle (ΔT) should be correlated to B/Ca and temperature itself not? This needs further explanation.*

   **Response:** We decided to focus on ΔT, which was calculated as the difference between the maximum and minimum seawater temperature registered during the algal growth. The definition of ΔT and its meaning has been added in Materials and Methods section. As mentioned above, we believe that this measure better characterizes the sampling sites compared to absolute temperature, given their differences in depth and geographical regions. Elba and Pontian Isl., for example, have both a temperature mean of 15°C. Nevertheless, in Pontian Isl. (66 m depth), the temperature keeps more constant throughout the period (ΔT = 3°C), while in Elba there is a higher amplitude of temperature variations (ΔT = 5°C). The effects of these fluctuations around the optimum of algal growth would not have been highlighted by the mean temperature value. Last, but not least, the resolution of the laser ablation does not allow us to precisely discriminate the month of year the analysis is referring to. Therefore, we cannot attribute a single point of analysis to an absolute temperature in a specific time of the year, but rather refer more generically to the cold and warm season. In fact, dark and light bands are usually associated to cold and warm periods and we have used this estimation to create the plot you suggested (Figure 12). In this plot the maximum and minimum temperature of the year corresponds to light and dark bands and the periodical oscillations in Mg, Li and Sr/Ca reveal their reliability as temperature proxies. Nevertheless, as strongly recommended by all the Referees, we decided to change Figure 9 in the revised text (Figure 11), in order to show more clearly the relationship between B/Ca and seawater temperatures. We therefore plotted also seawater maximum and minimum temperature values with B/Ca mean values in long and short cells, respectively. As you can see in the resulting Figure 11, temperature shows a very poor influence on B data.

- **Comment:** *Line 350: here a "poor correlation with temperature" is mention but it is a poor correlation with the magnitude of the seasonal temperature change that is plotted in fig9. Please change wording here.*

   **Response:** The Figure 9 (Figure 11) has been changed as mentioned in the previous response.

- **Comment:** *Line 354: This ending to the discussion unfairly represents the empirical calibration work that has gone into calibrating the δ11B proxy in CA (e.g. Anagnostou et al., 2019). The controls on B/Ca in corals and foraminifera are also less well known compared to the strong seawater pH control on the δ11B of these carbonates and therefore the use of δ11B should not be discouraged for reconstructing past seawater pH*

   **Response:** Our comment was referring more specifically to the B/Ca; the original text was ambiguous, and it was therefore reworded.

- **Comment:** *Line 363: There is no mention of Mg/Ca relationship in with deltaT in the main text, so it is strange to see this in the conclusions. It is again unclear why correlation to seasonal temperature range is important particularly when average Mg/Ca appears to correlate poorly with average temperature at these sites.*

   **Response:** The text has been modified for clarity. There is indeed no reference to a correlation in the main text, but rather a discussion about the fact that Mg/Ca means in the different sites mirror the

amplitude of seawater temperature variations during the algal growth. The explanation of the importance of ΔT is in the comment above.

**References**

Adey, W. H.: The genus *Clathromorphum* (Corallinaceae) in the Gulf of Maine, Hydrobiologia, 26, 539–573, doi:10.1007/BF00045545, 1965.

Anagnostou, E., Williams, B., Westfield, I., Foster, G. L., and Ries, J. B.: Calibration of the pH-$\delta^{11}$B and temperature-Mg/Li proxies in the long-lived high-latitude crustose coralline red alga *Clathromorphum compactum* via controlled laboratory experiments, Geochim. Cosmochim. Acta, 254, 142–155, doi:10.1016/j.gca.2019.03.015, 2019.

Caragnano, A., Basso, D., Jacob, D. E., Storz, D., Rodondi, G., Benzoni, F. and Dutrieux, E.: Coralline red alga *Lithophyllum kotschyanum* f. affine as proxy of climate variability in the Yemen coast, Gulf of Aden (NW Indian Ocean), Geochim. Cosmochim. Acta, 124, 1–17, doi:10.1016/j.gca.2013.09.021, 2014.

Caragnano, A., Basso, D., Storz, D., Jacob, D. E., Ragazzola, F., Benzoni, F. and Dutrieux, E.: Elemental variability in the coralline alga *Lithophyllum yemenense* as an archive of past climate in the Gulf of Aden (NW Indian Ocean), J. Phycol., 53, 381–395, doi:10.1111/jpy.12509, 2017.

Donald, H. K., Ries, J. B., Stewart, J. A., Fowell, S. E. and Foster, G. L.: Boron isotope sensitivity to seawater pH change in a species of *Neogoniolithon* coralline red alga, Geochim. Cosmochim. Acta, 217, 240–253, doi:10.1016/j.gca.2017.08.021, 2017.

Frantz, B.R., Foster, M.S. and Riosmena-Rodríguez, R.: *Clathromorphum nereostratum* (Corallinales, Rhodophyta): The oldest alga?, J. Phycology, 41, 770–773, doi:10.1111/j.1529-8817.2005.00107.x, 2005.

Halfar, J., Steneck, R. S., Joachimski, M., Kronz, A. and Wanamaker, A. D. Jr.: Coralline red algae as high-resolution climate recorders, Geology, 36, 463–466, doi:10.1130/G24635A.1, 2008.

**Anonymous Referee #2**

Dear Referee #2,

Thank you for your comments. Our paper has been implemented and improved following your suggestions. English has also been checked throughout the text. We also appreciate that you have recognized the importance and novelty of our results.

Hereafter I will reply to each of your comments.

- **Comment:** *The authors use samples collected from different time intervals (1990–2017), but does not discuss how global warming changed water temperature and how much this consideration influence their discussion. Similarly, they discuss relationship between B/Ca and pH/DIC, but correction due to anthropogenic CO2 invasion (ocean acidification) is not considered.*

    **Response:** The point that you mentioned is certainly interesting but goes far beyond the objectives of this work and its methodological ability. The error attributable to the offsets between the proxy and the actual temperature/pH would probably be higher than the magnitude of variations occurred in that period because of climate change.

- **Comment:** *There is no explanation about analytical precision of LA-ICPMS used in this study. I wonder how reliable and reproducible their analysis is?*

    **Response:** More details regarding the analytical precision of the method have been added in the Section Materials and Methods. Accuracy and precision were higher than 4% and 8% for NIST 612 and Ca standards respectively.

- **Comment:** *There is no figure on Mg/Ca variation and its comparison to dark/light growth bands for each sample, which likely aids readers to understand the discussion. For example, the authors argue that the Aegadian sample records 10 years of growth, but the reader cannot see this.*

    **Response:** A figure showing Mg and other elements oscillations across the algal thallus of the sample collected in Morlaix has been added (Figure 12), evidencing light and dark growth bands variations. A new figure showing the laser transects crossing in longitudinal section the samples collected in the Mediterranean sites will be added as .pdf Supplement (Figure S1), if the editor agrees.

- **Comment:** *The authors argue that Li/Ca and Sr/Ca of CA are temperature proxy, because there is linear relationships between Li/Ca and Mg/Ca as well as between Sr/Ca and Mg/Ca, but I don't agree this. There is no figure supporting a significant correlation between Mg/Ca and temperature. The relationship between Mg/Ca and temperature is a key of the discussions of this manuscript. A box plot in Fig. 4 shows no significant differences among sites (N = 4). Also, there could be a pseudo-correlation between Li/Ca and Sr/Ca with temperature, because many environmental factors vary in the same phasing (e.g., irradiance, temperature, growth rate).*

    **Response:** The reliability of Mg/Ca as a temperature proxy in calcareous red algae has been recognized since the mid-1990s (Chave and Wheeler, 1965; Moberly, 1968; Henrich et al., 1996; Halfar et al., 2000; Kamenos et al., 2008). We do not have evidence to cast doubts on this hypothesis. On the contrary, the plot we added in the revised version (Figure 12), as you and the other reviewers suggested, clearly shows the relationship between the seasonal oscillations in Mg, Li and Sr/Ca ratios and the seawater temperature, across light and dark bands. Moreover, Mg/Ca mean values across sites, although they are not statistically significant, decrease as the magnitude of temperature fluctuations during the algal growth decreases. What you were referring to as a "pseudo-correlation" could be the case of any

element incorporated by the alga, it is true. The limit of performing geochemical analyses on wild-grown algae is the impossibility to control all the environmental factors during algal growth, contrary to what is done during culture experiments. On the other hand, recognizing patterns in wild-grown specimens is only possible when, despite variability, an overarching control can be observed, which is the strength and the novelty of our observations.

- **Comment:** *"p" of pKb and pCO2 should be italic.*

  **Response:** The text has been corrected as you suggested.

- **Comment:** *Line 32: What is "normal pH conditions"?*

  **Response:** The text has been changed to "typical surface seawater conditions".

- **Comment:** *Line 35–36: The notion that "Seawater isotopic composition δ11Bsw is 39.61‰ (Foster et al., 2010) and varies with the isotopic composition of B(OH)3 and B(OH)4-" depend on timescales concerned.*

  **Response:** In the text it has been specified that we were referring to current seawater isotopic composition (Foster et al., 2010).

- **Comment:** *Line 41: The expression that TA and DIC "are closely related to the δ11B of the borate" is wired. Yes, DIC and TA is related to pH, and pH can be indirectly estimated from d11B of some kind of calcium carbonate.*

  **Response:** Line 40: "The $p$CO$_2$, the seawater [CO$_3^{2-}$], pH and [B(OH)$_4^-$] are mainly controlled by the balance between the total alkalinity (TA) and the dissolved inorganic carbon (DIC) and are closely related to the $\delta^{11}$B of the borate". The subject of this sentence was "$p$CO$_2$, the seawater [CO$_3^{2-}$], pH and [B(OH)$_4^-$]", not TA and DIC. However, the text has been modified in the revised version for clarity.

- **Comment:** *The notion that "the boron-to-calcium ratio (B/Ca) is informative about past seawater CO32- concentrations (Yu and Elderfield, 2007; Yu et al., 2007; Rae et al., 2011)" is only true to foraminifera. As for reef building corals, B/Ca is related to other carbonate parameters (see Holcomb et al., 2016).*
  *Holcomb, M., DeCarlo, T. M., Gaetani, G. A., & McCulloch, M. (2016). Factors affecting B/Ca ratios in synthetic aragonite. Chemical Geology, 437, 67–76. https://doi.org/10.1016/j.chemgeo.2016.05.007*

  **Response:** Thank you for the reference. The relationship between B/Ca and CO$_3^{2-}$ had been tested empirically in foraminifera, as you pointed. Holcomb et al. (2016) performed synthetic aragonite precipitation experiments, also confirming the suitability of B/Ca as a proxy of [CO$_3^{2-}$]. Scarce data exist on coralline algae, which are composed by high-Mg calcite, and there is no specific mention about B/Ca coupled with CO$_3^{2-}$ data. This has been specified in the text. In culture experiments of the coralline alga *Neogoniolithon* sp., Donald et al. (2017) found a negative effect of DIC on B/Ca.

- **Comment:** *Line 58: What is "indeterminate growth". Please be more quantitative. Also, "no ontogenetic trend" of what?*

**Response:** The concept has been clarified in the text. The term "indeterminate growth" is commonly used when referring to the longevity of coralline algae (Adey, 1965; Frantz et al., 2005; Halfar et al., 2008). Their growth trend does not slow down asymptotically with age, following the so-called "ontogenetic trend" as bivalves do. Coralline algae thus preserve the resolution of the geochemical signals even in the later stages of growth.

- **Comment:** *Lines 70–73: I could not understand this line "Achieving the best reliability of geochemical proxies for climate reconstructions is indeed crucial, which drives a growing interest on multiple approaches, by considering multi-proxies for a single environmental factor (D'Olivo et al., 2018; Zinke et al., 2019; Cuny- Guirrec et al., 2019), as well as the influence of multi-factors on a single proxy (Kaczmarek et al., 2016; Donald et al., 2017)."*

  **Response:** The text has been rephrased for clarity, as following: "To achieve the best reliability of geochemical proxies for climate reconstructions, a growing interest has arisen on multiple approaches, by considering multi-proxies for a single environmental factor (D'Olivo et al., 2018; Zinke et al., 2019; Cuny-Guirrec et al., 2019). Moreover, it is important to recognize the influence of multiple factors on a single proxy (Kaczmarek et al., 2016; Donald et al., 2017)."

- **Comment:** *Line 76: "positive correlations" between what?*

  **Response:** "positive correlations" has been deleted. Indeed, the correlations we were referring to are specified in the next sentence.

- **Comment:** *Line 77 "B" should be "B/Ca" or "[B]"?*

  **Response:** B/Ca, it has been changed in the text.

- **Comment:** *Line 79 "Foslie 1898" should be (Foslie)?*

  **Response:** *Clathromorphum compactum* (Kjellman) Foslie 1898 is the correct nomenclature. Please check Guiry & Guiry AlgaeBase (https://www.algaebase.org/). The use of parentheses has specific meaning in nomenclature.

- **Comment:** *Line 78: Is "sea surface temperature (SST)" relevant to this study? Temperature is better here.*

  **Response:** The text has been modified.

- **Comment:** *Line 83 About "calcifying species". Mg/Ca is a paleo-thermometer with regard to calcifying organisms that have calcite crystal form such as foraminifera and CA. Sr/Ca, not Mg/Ca, is paleo-thermometer in coral skeleton.*

  **Response:** We agree with your comment. The papers cited in the sentence refer to coralline algae and we have written explicitly "calcite lattice".

- **Comment:** *Line 82–87: I wonder why the authors don't mention Li/Mg temperature proxy (especially for coral) here, because which is commonly used.*
  *K. Cuny-Guirriec, E. Douville, S. Reynaud, et al., Coral Li/ Mg thermometry: Caveats and constraints, Chemical Geology, Volume 523, 30 September 2019, Pages 162-178*

**Response:** Mg/Li calibration did not improve the Mg/Ca or Li/Ca temperature relationship in previous empirical studies on coralline algae (Caragnano et al., 2014; 2017), as for corals instead. To test this proxy, we added Mg/Li results in the revised text. We plotted Mg/Li data with the results from Anagnostou et al. (2019) on cultured coralline algae *Clathromorphum compactum* (Figure 6). Moreover, we also plotted Mg/Li oscillations through time, which showed a poor relationship with seawater temperature (Figure 7), contrary to the other temperature proxies (Figure 12).

- **Comment:** *Lines 92–95: Please revise these sentences: "In this paper, we measure by LA-ICP-MS for the first time the temperature proxies (Mg/Ca, Sr/Ca, Li/Ca) and B/Ca in the nongeniculated CA Lithothamnion corallioides (P. Crouan & H. Crouan) P. Crouan & H. Crouan 1867 collected from different geographic settings and depths across the Mediterranean Sea and in the Atlantic Ocean."*

  **Response:** We have revised the sentence.

- **Comment:** *Line 97: Remove "Here, "*

  **Response:** The text has been modified.

- **Comment:** *Line 100: Please revise these sentences: "In this paper, we test the B/Ca ratio versus the temperature proxies and the growth rates in order to evaluate their effects on B incorporation, which, indeed, could distort the B signal used for paleoclimate reconstructions."*

  **Response:** The sentences have been revised.

- **Comment:** *Line 116–122: Please revise these sentences: "Morphological identification was based on Adey & McKibbin (1970), Irvine & Chamberlain (1994), and other information about maerl species distribution provided by Carro et al. (2014) and Melbourne et al. (2017). The samples selection started from a much wider collection than the one eventually selected for the chemical analyses. Particularly, the Atlantic sample (Morlaix) was used as voucher specimen for the subsequent identification of the Mediterranean samples, since Phymatolithon spp. and L. corallioides are the major components in the Atlantic maerl (Hall-Spencer et al. 2010; Carro et al., 2014). Hence, once excluded the belonging to the genus Phymatolithon, the Morlaix sample identified as L. corallioides was used as a reference for the most reliable identification of the other Mediterranean samples."*

  **Response:** The sentences have been revised.

- **Comment:** *Line 124: How large "algal branches" were? I guess less than 5 mm from Fig. 2.*

  **Response:** Correct, the algal branches were to the order of 5 mm as shown in Figure 2.

- **Comment:** *Line 125 What is "treated samples"? Does this mean that the samples were maintained for 24h in a resin under the vacuum (drying)?*

**Response:** Correct, the treated samples are the included algal branches. The text has been modified for more clarity.

- **Comment:** *Line 132: Should be "according to", instead of "in agreement with"?*

  **Response:** The text has been changed.

- **Comment:** *Line 130 (Section 2.3) I wonder the authors did pre-ablation of CA sample surface? The usage of distilled water instead of MQ water is enough to remove surface contamination? Especially, boron is easily contaminated from the environment.*

  **Response:** We did not perform pre-ablation. However, during processing the signal was excluded in the first part, as potentially affected by contamination. This has been specified in the Materials and Methods section.

- **Comment:** *Lines 144–149: Please revise these sentences: "In the absence of in-situ environmental data, the seawater temperature data have been extracted by at least 11 years of monthly reanalysis spanning 1980-2017 from ORAS5 (Ocean ReAnalysis System 5), at 0.25-degree horizontal resolution (Zuo et al., 2019). The nearest sea point of the three-dimensional numerical grid was considered for each sample location. Details of the time interval considered for each sampling site are shown in Table 2. Minimum, maximum and mean values, as reported in Table 2, refer to the temperature at sampling depth and have been measured on the entire time interval for which the data have been extracted"*

  **Response:** The sentences have been revised.

- **Comment:** *Lines 150–158: Please revise these sentences: Carbon data in each sampling site have also been extracted. They were not available in the same time interval of temperature data. Nevertheless, the seasonal variations occurring in the extracted period have allowed the characterization of the sampling sites. Monthly mean seawater pH has been derived by the CMEMS (E.U. Copernicus Marine Service Information) global biogeochemical hindcast spanning 1993-2018, at 0.25-degree horizontal resolution. Monthly means of DIC in 2019 and 2020 have been extracted by CMEMS biogeochemical analysis and forecasts for the Mediterranean Sea, at 0.042-degree horizontal resolution (Salon et al., 2019; Bolzon et al., 2020). In the Atlantic site, monthly means of DIC were derived from CMEMS IBI biogeochemical forecasts, at 0.028-degree horizontal resolution covering the years 2019-2020. Minimum, maximum and mean values of DIC, as reported in Table 2, refer to sampling depth and have been measured on the entire time interval of extraction."*
  *Also, I have a great concern here. Surface seawater DIC is changing over time due to CO2 invasion from the atmosphere. Thus, the authors need to correct this influence when comparing pH and DIC among sampling sites (Lines 189–192).*

  **Response:** The sentences have been revised. pH and DIC data refer to the sampling depth in each site (12, 40, 45 and 66 m depth), not to the surface. We do not believe that the influence of anthropogenic $CO_2$ could be relevant to our analyses, due to the resolution of the method as explained above for climate change issues.

- **Comment:** *Line 191 and Table 2: An more common unit of DIC is µmol/kg, not mol/m3.*

**Response:** The unit has been converted, thank you for the suggestion.

- **Comment:** *Line 160–161: I wonder why they made estimation of linear growth rate in this m manner, because this can be done by using image software such as Image J using Fig. 2b. Is it due to the fact that CA has many "faint bands" and variation of Mg/Ca is more reliable to see summer/winter seasonality?*

**Response:** Correct, the estimation of growth rates under light microscope allowed us to distinguish seasonal bands with a higher degree of confidence, since these bands are not obvious in the images (Figure 2 and Figure S1). Moreover, once confirmed the positive correlation between temperature and Mg/Ca variations, Mg/Ca peaks were used to double-check the correspondence of the years, improving reliability.

- **Comment:** *Line 179: Please explain what "ΔT" stands for in the main text, not in the Table caption.*

**Response:** The definition of ΔT has been added in the Materials and Methods (Section 2.3). Thank you for your comment.

- **Comment:** *Lines 181–189: Does the second decimal place in water temperature have any meaning?*

**Response:** The approximation has been changed to one decimal place.

- **Comment:** *Line 208 The word "widest oscillation" sounds wired. Please rephrase it.*

**Response:** The sentence has been rephrased.

- **Comment:** *Lines 216–217: It's a circular argument, because the authors use variation of Mg/Ca to distinguish light and dark bands.*

**Response:** The description of the method we used to distinguish between dark and light bands has been implemented, because the text was probably misleading. In order to make a reliable distinction between long and short cells, we discarded intermediate Mg/Ca values that probably refer to middle seasons, as well as data from spots positioned on faint bands.

- **Comment:** *Lines 233–234: Is there any statistics showing no correlation between B/Ca and temperature?*

**Response:** B/Ca in the Mediterranean samples had no significant correlations with temperature proxies Mg, Li and Sr/Ca, resulting in a Spearman's p value > 0.05, as cited in the text. This is also true for the correlation between B/Ca and seawater temperatures in the different sites. We changed Figure 9 in the revised text (Figure 11), in order to show more clearly the relationship between B/Ca and seawater temperatures. We therefore plotted also the maximum and minimum temperature values registered in the extracted time interval with B/Ca mean values in long and short cells, respectively. As you can see in the resulting Figure 11, temperature shows a very poor influence on B data. Moreover, Figure 12 shows a poor relationship between B/Ca oscillations and seasonal temperature variations even in Morlaix, contrary to what observed for Mg, Li and Sr/Ca.

- **Comment:** *Line 271–272: Positively correlated, but insignificant, right? (Probably due to a small sample number: N = 4)*

**Response:** Yes, the correlation is strongly positive with a "borderline" p value of 0.051. As you wrote, this could be related to the small sample number, but we believe it is noteworthy.

- *Comment: Lines 288–295: I could not understand this lines. How important the B/Ca range comparison between cultured and naturally obtained (wild) CAs is? There is no mention about correlation between B/Ca and pH/pCO2/DIC, etc.*

**Response:** What we want to stress is precisely the paucity of data on B/Ca in coralline algae. As you pointed, the papers we cited to compare our data with the literature refer to cultured algae. Using LA-ICP-MS could probably have contributed to the wider range of B/Ca measured in *L. corallioides*, compared to the bulk analyses performed by Donald et al. (2017) and Anagnostou et al. (2019), as suggested by Referee#1 and specified in the text. However, there are no other data on wild-grown coralline algae collected across sites with different environmental settings, neither other geochemical analyses on *L. corallioides*. The correlation between B/Ca and carbonate system parameters is discussed in the following paragraphs (Line 317) and has been further implemented.

- *Comment: Line 297: What does "the preservation state of mineral structures" influence trace element/Ca ratio of CA? Does this mean that they can be altered in a live-caught specimen?*

**Response:** Early diagenesis during growth stages in rhodoliths could compromise the geochemical signals and therefore a prior evaluation of the conservation status of the alga should be important.

- *Comment: Lines 299–301: I'm not convinced by this argument, because there is no figure supporting a significant correlation between Mg/Ca and temperature. A box plot in Fig. 4 shows no significant differences among sites. Based on a correlation of Mg/Ca with temperature (not shown), the authors argue that "For the first time, we confirmed here the reliability of the temperature proxies Li/Ca and Sr/Ca on a deep-water Mediterranean CA"? I wonder how reliable these proxies as thermometer are. Similarly, there is no evidence that "The results of the statistical analyses on Mg/Ca evidenced a strong relationship with the seawater temperatures extracted from ORAS5 (Table 2), as expected."*

**Response:** Mg/Ca has successfully been applied as temperature proxy in coralline algae and we have no reason to doubt its reliability. The revised paper also includes a new plot showing the cyclic variations of Mg, Li and Sr/Ca through time which are evidently related to seasonal temperature variations (Figure 12). Moreover, as responded in the comments above, even if the differences in Mg/Ca shown in Figure 4 are not statistically significant, Mg/Ca mean values decrease as the magnitude of temperature fluctuations during the algal growth decreases.

- *Comment: Lines 311–313: There could be a pseudo-correlation between Li/Ca and Sr/Ca with temperature, because many environmental factors vary in the same phasing (e.g., irradiance, temperature, growth rate). Classically, Mg/Ca of reef-building corals had been regarded as temperature proxy (now we know it is not). See, for example: Inoue M., Suzuki A., Nohara M., Hibino K. and Kawahata H. (2007) Empirical assessment of coral Sr/Ca and Mg/Ca ratios as climate proxies using colonies grown at different temperatures. Geoph. Res. Lett. 34, L1261. https://doi.org/10.1029/ 2007GL029628.*
*B/Ca is also well co-related with Mg/Ca (Fig. 6), but not regarded as temperature proxy, according to the authors.*

**Response:** As we already discussed in the major point above, we cannot exclude the influence of multiple factors on elements incorporation, which is likely indeed. Our samples belong to calcareous red algae (coralline algae) that have obvious major differences with the physiology and mineralogy of corals. They have been collected in nature and therefore we could not control all the environmental factors involved during algal growth. Nevertheless, Li and Sr/Ca correlation with temperature is evidenced in the new plot showing seasonal oscillations across light and dark bands (Figure 12). B/Ca results instead could not support its application as temperature proxy because its correlation with Mg/Ca is only significant in the sample from Morlaix, the shallowest site, highly influenced by seasonality (as shown by the highest temperature fluctuations). Also, in Morlaix B/Ca variations during the algal growth did not reflect the seasonal temperature changes (Figure 12), contrarily to the proper temperature proxies.

- **Comment:** *Line 321: 8.32!?*

  **Response:** The text has been corrected.

- **Comment:** *Line 326: pH and DIC "results"? The authors did not analyze seawater but just extracted data from the database.*

  **Response:** The text has been changed referring specifically to extracted data.

- **Comment:** *Line 351: The title of this paper is based on the fact that there is "a closer relationship with growth rate rather than temperature" even though there is no statistical significance?*

  **Response:** Figure 9 (Figure 11) shows a noteworthy positive relationship between B/Ca and growth rates. We suggest that the different growth rates could have contributed to the B/Ca differences among sites that could not be justified by the differences in carbonate system parameters and are not related to temperature either.

- **Comment:** *Line 356: Does the authors would like to mention possibility of local modification of boron isotopic composition of seawater?*

  **Response:** Foster et al. (2010) measured a mean seawater $\delta^{11}B$ of 39.61 ‰ from several ocean basins and found no significant or systematic variations of $\delta^{11}B$ with depth, salinity, or temperature, in agreement with its long residence time in seawater and despite the large range in salinity (32/38 psu) and temperature (-0.3/25.9 °C) considered. We therefore have no reason to believe in a modification of seawater boron isotopic composition among our sampling sites.

**References**

Adey, W. H.: The genus *Clathromorphum* (Corallinaceae) in the Gulf of Maine, Hydrobiologia, 26, 539–573, doi:10.1007/BF00045545, 1965.

Anagnostou, E., Williams, B., Westfield, I., Foster, G. L., and Ries, J. B.: Calibration of the pH-$\delta^{11}B$ and temperature-Mg/Li proxies in the long-lived high-latitude crustose coralline red alga *Clathromorphum compactum* via controlled laboratory experiments, Geochim. Cosmochim. Acta, 254, 142–155, doi:10.1016/j.gca.2019.03.015, 2019.

Caragnano, A., Basso, D., Jacob, D. E., Storz, D., Rodondi, G., Benzoni, F. and Dutrieux, E.: Coralline red alga *Lithophyllum kotschyanum* f. affine as proxy of climate variability in the Yemen coast, Gulf of Aden (NW Indian Ocean), Geochim. Cosmochim. Acta, 124, 1–17, doi:10.1016/j.gca.2013.09.021, 2014.

Caragnano, A., Basso, D., Storz, D., Jacob, D. E., Ragazzola, F., Benzoni, F. and Dutrieux, E.: Elemental variability in the coralline alga *Lithophyllum yemenense* as an archive of past climate in the Gulf of Aden (NW Indian Ocean), J. Phycol., 53, 381–395, doi:10.1111/jpy.12509, 2017.

Chave, K. E. and Wheeler, B. D.: Mineralogic changes during growth in red algae, *Clathromorphum compactum*, Science, 147, 1965.

Cuny-Guirriec, K., Douville, E., Reynaud, S., Allemand, D., Bordier, L., Canesi, M., Mazzoli, C., Taviani, M., Canese, S., McCulloch, M., Trotter, J., Rico-Esenaro, S. D., Sanchez-Cabeza, J.-A., Ruiz-Fernàndez, A. C., Carricart-Ganivet, J. P., Scott, P. M., Sadekov, A. and Montagna, P.: Coral Li/Mg thermometry: Caveats and constraints, Chem. Geol., 523, 162–178, doi:10.1016/j.chemgeo.2019.03.038, 2019.

D'Olivo, J. P., Sinclair, D. J., Rankenburg, K. and McCulloch, M. T.: A universal multi-trace element calibration for reconstructing sea surface temperatures from long-lived *Porites* corals: removing 'vital-effects', Geochim. Cosmochim. Acta 239, 109–135, doi:10.1016/j.gca.2018.07.035, 2018.

Donald, H. K., Ries, J. B., Stewart, J. A., Fowell, S. E. and Foster, G. L.: Boron isotope sensitivity to seawater pH change in a species of *Neogoniolithon* coralline red alga, Geochim. Cosmochim. Acta, 217, 240–253, doi:10.1016/j.gca.2017.08.021, 2017.

Foster, G. L., Pogge von Strandmann, P. A. E. and Rae, J. W. B.: Boron and magnesium isotopic composition of seawater, Geochem. Geophys., 11 (8), Q08015, doi:10.1029/2010GC003201, 2010.

Frantz, B.R., Foster, M.S. and Riosmena-Rodríguez, R.: *Clathromorphum nereostratum* (Corallinales, Rhodophyta): The oldest alga?, J. Phycology, 41, 770–773, doi:10.1111/j.1529-8817.2005.00107.x, 2005.

Halfar, J., Zack, T., Kronz, A. and Zachos, J. C.: Growth and high resolution palaeoenvironmental signals of rhodoliths (coralline red algae): a new biogenic archive, J. Geophys. Res. C, 105, 22107–22116, 2000.

Halfar, J., Steneck, R. S., Joachimski, M., Kronz, A. and Wanamaker, A. D. Jr.: Coralline red algae as high-resolution climate recorders, Geology, 36, 463–466, doi:10.1130/G24635A.1, 2008.

Henrich, R., Freiwald, A., Wehrmann, A., Schafer, P., Samtleben, C. and Zankl, H.: Nordic-cold water carbonates: occurrence and controls. In Reitner, J., Neuweiler, F. and Gunkel, F.: Global and Regional Controls on Biogenic Sedimentation (Eds.), Gottinger Arbeiten Geol. Palaeontol., Gottingen, 35–53, 1996.

Holcomb, M., DeCarlo, T. M., Gaetani, G. A., and McCulloch, M.: Factors affecting B/Ca ratios in synthetic aragonite, Chem. Geol., 437, 67–76, doi:10.1016/j.chemgeo.2016.05.007, 2016.

Kaczmarek, K., Nehrke, G., Misra, S., Bijma, J. and Elderfield, H.: Investigating the effects of growth rate and temperature on the B/Ca ratio and $\delta^{11}B$ during inorganic calcite formation, Chem. Geol., 421, 81–92, doi:10.1016/j.chemgeo.2015.12.002, 2016.

Kamenos, N. A., Cusack, M. and Moore, P. G.: Coralline algae are global paleothermometers with bi-weekly resolution, Geochim. Cosmochim. Acta, 72, 771–779, doi:10.1016/j.gca.2007.11.019, 2008.

Moberly, R.: Composition of magnesian calcites of algae and pelecypods by electron microprobe analysis, Sedimentology, 11: 61–82, doi:10.1111/j.1365-3091.1968.tb00841.x, 1968.

Zinke, J., D'Olivo, J. P., Gey, C. J., McCulloch, M. T., Bruggemann, J. H., Lough, J. M. and Guillaume, M. M. M.: Multi-trace-element sea surface temperature coral re-construction for the southern Mozambique Channel reveals teleconnections with the tropical Atlantic, Biogeosciences, 16, 695–712, doi:10.5194/bg-16-695-2019, 2019.

**Anonymous Referee #3**

Dear Referee #3,

Thank you for your review. Our revised paper has been implemented following your suggestions and raw data have been included. Thank you also for your comments on the value of our results.

Hereafter I will reply to each of your comments.

- **Comment:** *Environmental data. The environmental data the authors use for the collected samples does not correspond with the date of collection, many of the samples have been compared with averages of temperature and pH for years after their collection dates:*
  *Aegadian Island – Collected 1993, temperature 1986 to 2017, pH 1993 to 2018, DIC 2019-2020*
  *Elba Island – Collected 1990, temperature 1980 -1991, pH 1993 to 2018, DIC 2019-2020*
  *Morlaix – Collected 1991, temperature 1980-1992, pH 1993 to 2018, DIC 2019-2020*
  *Pontian Island – Collected 2017, temperature 1986-2017, pH 1993 to 2018, DIC 2019-2020*
  *I understand the pH and DIC data are sparse, but there doesn't seem to have been any attempt to try and 'translate' these values to the collection time, or discuss the implications (besides lines 150-152).*

  **Response:** Temperature data have been extracted by 11 years of monthly reanalysis for each sampling site in the revised paper (the time interval considered for Aegadian Isl. has been corrected). The selected time interval (11 years) approximately covered the period of algal growth crossed by the laser transects and was the same for each site, for comparability. This choice has been better explained in the revised text (Section 2.3). As you pointed, carbonate system data are sparse instead, because of their reduced availability. The selected time interval of extraction (2019-2020) was the only period available for each sampling sites. We used two different biogeochemical models to extract pH and DIC data from the sampling sites, as described in the Section 2.3. pH data have been derived by CMEMS global biogeochemical hindcast. DIC data, which largely dictate the pH, have been extracted from CMEMS biogeochemical forecasts in the Mediterranean Basin and in the Atlantic Ocean. Even using different models, both extractions showed the same variations among sites, with the lowest values of both pH and DIC in Morlaix, and higher values in the Mediterranean sites (Aegadian Isl., Pontian Isl. and Elba). This suggests that the extracted data are not model's artefacts and allowed us to characterize the sampling sites. In conclusion, carbon data were used to characterize the different sites, allowing us to discriminate between the influence of carbon system parameters and other factors on B/Ca. Temperature data, instead, allowed us to reconstruct the timeline of algal growth. The text has been implemented in the revised version.

- **Comment:** *'Raw' data. Like the other reviewers I miss the data of the individual samples and therefore it is difficult to make a fair assessment of many of the points raised by the authors. I don't know how many samples were actually analysed by the authors, is it one per site, or multiple. Likewise, how many measurements per sample. I would like to see the different elemental ratios per sample per site plotted, highlighting the measurements that represent the long or short cells. I would also plot the growth rate for each band (so that would be equal to length) as a shaded grey line behind.*

  **Response:** Raw data have been submitted to the PANGAEA repository. Moreover, a new table (Table 2) has been created to summarize the results of all element analyses in every sample. The analyses have been performed on one algal branch per site, as more clearly specified in the Materials and Methods section of the revised text. The laser transects consisted in 21-49 spots, which will be shown in the new supplement figure (Figure S1), if the Editor agrees. The Figure 12 in the revised paper shows the elemental ratio variations across the algal thallus of the Morlaix sample, as well as temperature changes through warm and cold seasons. Light and dark bands have also been evidenced.

- **Comment:** *Growth rate vs DeltaT: Figure 9 seems to be the justification for the much of the title and abstract. I have no problem with four datapoints, if those data points incorporated error bars the showed the full range of the B/Ca. I do however wonder what DeltaT is supposed to represent as a variable, are the authors implying that seasonality controls the absolute value of the B/Ca? That is not testing temperature as the title suggests but the seasonal range in temperature (the datapoint at DeltaT = 9 would be the lowest average temperature site). I note that range of the B/Ca (from the boxplots in figure 8) also eclipses the y-axis of figure 9, the intra sample/site ranges from ~300 to ~950 B/Ca. DeltaT. The authors use the range of temperature throughout and I am at a loss as to understand why you would compare a variable that is supposed to be a temperature proxy with the seasonality of the site. I could understand if the authors took the difference between max and min of the Element/Ca and compared it against DeltaT, but essentially the authors are comparing an absolute temperature with its supposed range.*

  **Response:** Error bars have been added to Figure 9 (Figure 11). As explained also in the comments to the other reviewers, we believe that the seasonal temperature change (ΔT), which was calculated as the difference between the maximum and minimum seawater temperature registered during the algal growth, better characterized the sampling sites compared to absolute temperature, given their differences in depth and geographical regions. As an example, Elba and Pontian Isl. have temperature means of 15°C. Nevertheless, in Pontian Isl. (66 m depth), the temperature keeps more constant throughout the period (ΔT = 3°C), while in Elba there is a higher amplitude of temperature variations (ΔT = 5°C). The effects of these fluctuations around the optimum of algal growth would not have been highlighted by the mean temperature value. Also, the laser ablation resolution does not allow us to precisely discriminate the month of year the analysis is referring to. Therefore, we cannot attribute a single point of analysis to an absolute temperature in a specific time of the year, but rather refer to the cold and warm season, as done for the creation of Figure 12, as suggested. For this reason, as strongly recommended by all the Referees, we decided to change Figure 9 (now Figure 11), to show more clearly the relationship between B/Ca and seawater temperature. We therefore plotted also the maximum and minimum temperature values per site with B/Ca mean values in long and short cells, respectively. As you can see in the resulting Figure 11, the poor relationship between temperature and B/Ca data is even clearer.

- **Comment:** *Long and short cells. The authors use the Mg/Ca to check if the data is from long and short cells, does this not make Figure 5 completely irrelevant, i.e., the difference between the long and short is because they were split by Mg/Ca ('Long cells were identified as spots with high Mg/Ca and positioning on light growth bands. Conversely, short cells were located in dark growth bands and corresponded to low Mg/Ca ratio.'). The all Mg/Ca (i.e., long+short) vs sites shows (Figure 4) no difference, therefore the only rationale to show figure 5 would be to inform the reader what cut off values you used to define short and long cells.*

  **Response:** The description of the method we used to distinguish between dark and light bands has been implemented for clarity. To make a reasonable distinction between long and short cells, we did not consider the data from the laser spots that had a doubtful interpretation. We thus discarded the data derived from spots positioned on faint bands and those referring to middle seasons, which have intermediate Mg/Ca values. By doing so, we preserved only the unequivocable information without any data manipulation.

- **Comment:** *Sampling at growth bands. Line's 165 and 135 'each growth band change which marked the transition between the cells usually produced in the warm season and those usually produced in the cold season,' you are targeting transitions between Cold and Warm*

*Seasons. So does the growth band really represent the cold season or warm season in its entirety? If you sample the transition between cold and warm by definition you don't sample the extremes (which DeltaT would represent) but do you fully sample the temperature? If a growth band represents a season, comparing the average of X number of growth bands is not entirely comparable with the mean temperature of the site (you could sample 3 cold and 4 warm growth bands, i.e., W-C-W-C-W-C-W, and therefore your average element/ca signal would be biased).*

**Response:** Referring to your first observation, the longitudinal sections were performed along the maximum branch thickness. As you can see in Figure 2, we targeted the centre of each bands which ideally correspond to the peak of temperature registered in the warm/cold period. We thus have tried to avoid sampling the transition between adjacent bands, as far as possible by the method. As explained in the previous response, when comparing short and long cells the information coming from doubtful points of analyses have been discarded.
Concerning your second observation, we performed the statistical analyses on both the original dataset and a modified dataset, which included the results from the same number of light and dark bands. This corrected dataset had the same results in the statistical tests and did not highlight differences or other additional information.

- **Comment:** *Probably avoid abbreviating coralline algae as CA if you're talking about [Element]/Ca (e.g., line 13 'Mg, Sr, Li and B in the CA')*

  **Response:** The text has been changed.

- **Comment:** *Line 21: "These pieces of evidence suggest that growth rates, triggered by the different ΔT and light availability across depth, affect the B incorporation in L. corallioides." – does this not contradict your title?*

  **Response:** Multiple factors act together in influencing the element incorporation of the algae during their growth. Nevertheless, according to our interpretation there is a major control of growth rates, rather than temperature, on B/Ca. For example, in Morlaix, at very shallow depth (12 m), the apparent influence of temperature on B/Ca is more likely linked to growth rate changes due to the high seasonality of the site (expressed by the high ΔT value).

- **Comment:** *Line 42: define acronym - MAS NMR*

  **Response:** The text has been changed.

- **Comment:** *Line 83: "in calcifying species, including CA (Barker et al., 2005;" – perhaps split the references between those that did and did not use coralline algae*

  **Response:** The text has been changed.

- **Comment:** *Line 92 and Line 100: A bit repetitive use of 'in this paper', perhaps at line 100 change to 'With our new measurements we aim to test whether…' to clarify objectives*

  **Response:** The text has been changed.

- **Comment:** *Line 105: If grab samples were used, how sure are you they were collected living specimens. Also in general, is collection date supposed to be read as collected living at this date*

  **Response:** Living and non-living samples were discriminated directly on board after grabbing, by some of the authors (DB and VB) using visual analysis. The red pigments of the algae rapidly degrade in

seawater after death, facilitating the distinction. The specimens used for this work were all alive at the time of collection.

- *Comment: Line 120: Missing word - 'Hence once excluded [specimens?] belonging to the genus…'*

**Response:** The text has been changed.

- *Comment: Line 114- 122: Clarify and expand 'how' (i.e., traits and characteristics used) and 'why' identification was needed, in the 'why' justify species selection.*

**Response:** Species identification was achieved by using classical morphometrical descriptors based on epithallial and perithallial cells observations on SEM images. Macromorphology, indeed, is not sufficient to discriminate between *P. calcareum* and *L. corallioides*. *L. corallioides* is a suitable species for the wide geographic scope of this work due to its presence in both Mediterranean and Atlantic waters. The text has been implemented in the revised paper.

- *Comment: Line 124: replace included with embedded/fixed/placed*

**Response:** The text has been changed.

- *Comment: Line 125: replace 'kept drying' with 'left to dry'*

**Response:** The text has been changed.

- *Comment: Line 125: 'included branches' – replace with 'treated branches'*

**Response:** The text has been changed.

- *Comment: Line 144 – 149: why 11 years?*

**Response:** The selection of this time interval has been explained in the main response above and has been implemented in the revised text.

- *Comment: Line 160: "The obtained values, expressed in linear extension over year (mm/yr), were crossreferenced with Mg/Ca results in order to check for the correspondence of Mg peaks with growth band helpful in highlighting faint bands and to achieve a more reliable estimate of the algal growth" – is this not circular? I note this is further expanded at lines 216-217 to 'Long cells were identified as spots with high Mg/Ca and positioning on light growth bands. Conversely, short cells were located in dark growth bands and corresponded to low Mg/Ca ratio'*

**Response:** The method used to discriminate between long and short cells has been implemented in the revised text. A detailed explanation is in the response to your main point above.

- *Comment: Line 163-165: what is the difference between the growth rates of the light and dark/long and short cells?*

**Response:** It was not possible to infer the growth rates of single bands with an acceptable error. We measured algal growth rates by dividing the laser transect to the number of years of growth. This method implies a non-negligible error margin and increasing in resolution as you suggest would increase the error as well and the data would lose reliability.

- **Comment:** *Line 200: "Both Li/Ca and Sr/Ca records had positive correlations with Mg/Ca in our samples" is this not because it's a closed sum? With an elemental ratio there is only so much subsitution available in the calcite lattice, so what do these correlations show?*

  **Response:** The correlations show a covariance between Li and Sr/Ca and Mg/Ca, which is mainly controlled by temperature. This suggest that also Li and Sr/Ca are temperature dependent. This is supported by Figure 12 in the revised text which shows the seasonal variations in these element ratios, related to the seasonal temperature changes. The same trend is not observed for B/Ca, in fact, the correlation between B/Ca and the temperature proxies is not supported.

- **Comment:** *Line 277: 'In order to provide high-resolution geochemical data on long and short cells separately, we considered only the results from the spots where the positive/negative Mg/Ca peaks correspond respectively to the light/dark bands resulting from the image analyses.' + Line 281-282: 'In the sample from Elba the growth bands were not clearly visible, preventing the analyses of trace elements on long and short cells separately.' These should occur earlier at sections 2.4 growth and 2.5 data.*

  **Response:** The text has been changed.

- **Comment:** *Line 299: Is 60m deepwater?*

  **Response:** The term "deep" has been corrected throughout the revised text.

- **Comment:** *Line 300: 'The results of the statistical analyses on Mg/Ca evidenced a strong relationship with the seawater temperatures extracted from ORAS5 (Table 2), as expected' – Table 2 is only the temperature so we as readers don't know if it is a strong relationship plus the extracted environmental variables in places do not correspond with the sampling dates. Plot the data not as a box plot (fig 4) but as a scatter plot with the full range of values as error bars.*

  **Response:** Concerning your first observation, the revised text has been implemented for clarity. We created the scatter plot you mentioned; however, we believe that plotting the data as a scatter plot would confound the reader and the data shown in Figure 4 are clearer as a box plot. Moreover, we performed a non-parametric test to evaluate the differences in Mg/Ca among sites, and the box plot showing the median values is more appropriate for these kind of statistics.

- **Comment:** *Line 307: 'Temperature covaries with irradiance and both correlate to seasons, which influence primary production, respiration and calcification in L. corallioides' - Do you (a) think that there is much irradiance change between 37 and 48N? There is roughly an 2 hr difference in length of day between 30 and 50 N. And (b) temperature lags irradiance, its why September is still relatively warm (as it takes time for the sea to 'warm up' due to seawaters heat capacity) along Europe's coasts.*

  **Response:** This sentence was cited from Martin et al. (2006). We understood your doubts about irradiance and the text has been changed for clarity. Indeed, the difference between Morlaix and the Mediterranean sites relies mainly on seasonal fluctuations in temperature (as highlighted by ΔT variations), and our discussion was not referring to irradiance.

- **Comment:** *Line 338: You can calculate the potential difference in light between 40 and 60 m to actually determine if irradiance matters, like so: The light at depth Z is equal to the light at surface multiplied by e to the power xz, where z is the depth and x is an extinction coefficient which varies from basin to basin.*

**Response:** This would be very interesting, and we thank you for your suggestion. Nevertheless, it would be beyond the aim of our present work.

- *Comment: Line 399: Data availability. Should be deposited in a repository rather than available on request (https://www.biogeosciences.net/policies/data_policy.html ).*

**Response:** Data have been submitted to the PANGAEA repository.

- *Comment: Line 414: Move 'Environmental data were provided by E.U. Copernicus Marine Service Information' to Data availability, include the url as well as the url of ORA S5.*

**Response:** The change has been made.

- *Comment: Figure 2B: The coralline algae is branching so I wonder how do you determine which is the oldest/youngest and therefore corresponds to your environmental data. For example, if you go up from between the 9th and 10th spots from the left, two branching growth lines meet, one sits slightly above the other (so that branch formed after). And because its branching does that not impact growth rate? I.e., what if the algae invested more time in branching than lengthening a branch?*

**Response:** Your observation is correct. The algae may grow in different directions and thus their growth patterns are often complicated. In this work, we followed one single growth direction per transect. The discontinuity you highlighted in Figure 2 marks a change in direction and of course the alga could have invested time in branching as you suggested. Our measurements of the growth rates, as responded above, have a certain margin of error which is also influenced by the complexity of the algal growth patterns. This is also the reason why we cannot reasonably measure the growth rates of single bands as mentioned above.

- *Comment: Figure 2B: Because the growth lines are convex how is growth measured, the longest point between two growth lines or fixed? Or more pertinently if the laser ablation track is used to define the size, how were the laser ablation sites positioned?*

**Response:** Longitudinal sections were cut across the maximum thickness of the branch. The laser transects crossed the growth bands by the mid length, as you can see in Figure 2.

- *Comment: Figure 5: If Mg/Ca is used to define the long and short cells this is not an independent test.*

**Response:** The method used to distinguish long cells from short cells has been implemented in the revised text, as explained in the major points above. Long and short cells are not dependent on Mg/Ca by definition, but they resulted to be so (Figure 5).

- *Comment: Figure 9: p = 0.051, technically that is above p <0.05.*

**Response:** Correct. We did not mention a statistically significant correlation given the borderline value, but the positive relationship shown in Figure 9 (Figure 11) is noteworthy.

- *Comment: Figure 9. The two lowest datapoints with B/Ca values of 460 and 610 look like they are subtly offset between the left and right panels*

**Response:** Figure 9 (now Figure 11) has been changed.

**References**

Martin, S., Castets, M.-D. and Clavier, J.: Primary production, respiration and calcification of the temperate free-living coralline alga Lithothamnion corallioides, Aquat. Bot., 85, 121–128, doi:10.1016/j.aquabot.2006.02.005, 2006.

---

## Author Response (AR2)

Dear Referee, thank you for your endeavour in revisioning the paper and for your suggestions. Some of your concerns have already been addressed in the previous review. Nevertheless, we made a further effort in clarifying those issues in the newly revised manuscript.

- **Comment:** Having read the previous reviews and the paper I think the paper needs to be considerably reworked. The concept certainly merits publication, but as per the previous reviews the raw data really needs to be shown, for all sites. The authors need to see it from the readers point of view, we are presented with lots of plots but not all of the data that we need to understand the authors assertion that growth rather than temperature controls B/Ca.
  **Response:** As already mentioned in the previous review, we agree on the importance of showing the raw data and, besides adding Table 2 in the previously revised version, we submitted the dataset in Pangaea, an open access repository. They are now publicly available at the following link: https://doi.pangaea.de/10.1594/PANGAEA.932201.

- **Comment:** Lots of the plots leave me with questions, why do I only get one out of 4 sites plotted for certain plots? Why is the entire sampling interval using in one panel and the third panel dividing the data between short and long cells (Figure 11)?
  **Response:** Most of the plots show the elemental analyses in the samples coming from all the sampling sites. Figure 8 shows the correlation plots between B/Ca and temperature proxies for the Morlaix sample only, because for the other samples there was a non-significant correlation (lines 252-255). The age model with the reconstruction of temperature variations during the algal growth has been presented for the Morlaix sample only. This choice has been made due to the meaningfulness of this sample for the scope of the paper. Indeed, besides having the best resolution of the growth bands among the samples, allowing the reconstruction of an age model reasonably accurate, the Morlaix sample was the only one where we found a correlation between temperature proxies and B/Ca. Since the purpose of the work was to investigate potential evidence of temperature and growth rate effects on B/Ca, we have welcomed the previous suggestions of the Referees to show the trace element variations at higher resolution, to check for seasonal temperature influences. We do not see the need to show the same for the Mediterranean samples in the present paper, since it would not add significant information to the conclusions. The manuscript has been implemented to justify this choice.
  Concerning your second question, as explained in the responses of the previous revision, and strongly recommended by all the Referees, we decided to modify Figure 11 to show more clearly the overall relationship between B/Ca and seawater temperature. Besides deltaT, which is meaningful for the comparisons of the sites at different depths, we therefore plotted the maximum and minimum temperature values per site with B/Ca mean values in long and short cells, respectively produced in the warm and cold periods. In the panel you see no relationship between B/Ca and the warm/cold season. The Morlaix sample also in this plot represents an exception that has been discussed in detail above. Acknowledging your comments, to make our use of temperature data clearer, we provided a more detailed explanation in materials and methods.

Major comments

- **Comment:** Why is only the Morlaix site chosen for plotting the comparisons between different variables and not the others? The authors need to show for all samples the transect of laser ablation values vs. length. I would envision this as a plot with several panels, first panel elemental ratio vs depth. Then a panel showing the position of the growth bands or at least where they

consider where the growth lines sit, overlaid with this the relative estimated growth rate for each band. That is the basic dataset, then plot as a scatter the data for each laser ablation point (i.e., the elemental value) vs growth rate, at the minute the authors are comparing the average growth rate over 8-11 years with the average B/Ca.

**Response:** Morlaix was chosen for the age model because of its significance to evaluate the temperature influences and because it was the one with the best visibility of the bands, as extensively explained in the previous response. As already commented in the first review (Referee#3), it was not possible to infer the growth rates of single bands with an acceptable error. The method used to measure growth rates, i.e. counting the number of growth bands crossed by the laser transect, implies a non-negligible error margin and increasing in resolution as you suggest would increase the error and the data would lose reliability. Although it would be very interesting to see these results, the approach you suggest is not feasible with these samples.

- **Comment:** Looking at figure 12 there is 'cyclical' nature of B/Ca and there is varying growth between the white and grey bands. Surely doing this would strengthen the authors argument that growth rate affects B/Ca?

  **Response:** We believe that the cyclicity you observe in B/Ca is more likely related to the differences in growth rates between cold and warm season, because of the high amplitude of seasonal fluctuations in temperature which characterize the site of Morlaix (highest deltaT), as we already discussed in the previous version of the paper (lines 342-346). White and grey bands are indicated to facilitate the interpretation of the plot, as done previously by other authors (e.g. Halfar et al., 2000); nevertheless, as you see in the image of the section, it does not strictly correspond to the width of growth bands. As already discussed, we are not able to define the different growth rates in each band with an acceptable approximation. As you might expect by looking at the thallus of *L. corallioides* in Fig. 1, the growth of this alga is not unidirectional neither perfectly regular. We cut the sample longitudinally along the expected growth direction of the branches, and we performed the analysis on the best resolved portion of the section. Nevertheless, the differences in the band width are not necessarily related to the growth rate but can also be due to changes in the growth direction.

- **Comment:** Plot all combinations of elemental ratios for all sites, as one figure per site, include this in the supplementary file if it's too many figures.

  **Response:** The complete dataset is available at Pangaea repository, as already mentioned. Moreover, in the previous revision we added Table 2 with the summary of all elemental ratios among sites. As you suggested, in the new revision we added the Figure S2 in the supplement also showing Li/Ca, Sr/Ca and Mg/Li among sites.

- **Comment:** Then the authors can assign the time interval for each ablation point. Rather than plotting min and max temperature for particular years (which led me to ask: "Why does time on the plots vary between February, March, August and September in Figure 7" as this is not referred to) just plot the full temperature time series.

  **Response:** The maximum and minimum values of Mg/Ca represent warm and cold periods of growth (Hetzinger et al., 2009), thus they should correspond to the highest and coldest seawater temperature. This method is widely used for the temperature calibration in coralline algae since it avoids uncertainties from sub-annual dating (Moberly, 1970; Corrège, 2006; Williams et al., 2014; Caragnano et al., 2014; Ragazzola et al., 2020). In the timeline reported in Fig. 12, the warmest

(August/September) and coldest (February/March) months of the temperature time series have been reported, as already wrote in the caption.

- **Comment:** Where are the growth bands of the Elba and Aegadian island? Having seen the supplement images I assume that the Mg/Ca is used almost exclusively for those samples. I note that the authors mention that such data is excluded, but I learn only at line 301 that means the entire sample for Elba. This should be discussed much earlier.
  **Response:** In the Elba sample the poor resolution of the growth bands did not allow us to clearly distinguish between the elemental signal in long and short cells. Thank you for your suggestion, this information has been also added in the Materials and Method section (2.5). As we already wrote in this section, the growth band counting was done directly under the light microscope, where it was much easier to see the bands by adjusting the light. The images in the Supplement, required by the previous Referee, were added to show the longitudinal section crossed by the laser transect. We still tried to adjust the image contrast and brightness in the revised Fig. S1 to better highlight the bands.

- **Comment:** Why gap fill the data in Figure's 7 and 12? We cannot then see which is the measured and which has been gap filled.
  **Response:** The image resolution during laser ablation forced us to take approximations while targeting the visible growth bands, therefore some of them may not have been targeted by the laser spot. Gaps have been highlighted by asterisks in the revised plots (Fig. 7, 12).

- **Comment:** Figure 11 shows there is no apparent relationship between temperature and B/Ca but then Figure 12 shows a similar pattern between B/Ca and temperature. So is the figure 11 not the results of comparing an average with an average? Considerable effort by the authors has gone into getting the ORAS5 temperature data, so I am at a loss as why they then bundle it up into a single average.
  **Response:** The laser ablation resolution does not allow us to precisely discriminate the month of year the analysis is referring to. Therefore, we cannot attribute a single point of analysis to an absolute temperature in a specific time of the year, but rather refer to the cold and warm season, as done for the creation of Fig. 7 and 12. This method is broadly used in literature to assess coralline algae age model (Moberly, 1970; Corrège, 2006; Williams et al., 2014; Caragnano et al., 2014; Ragazzola et al., 2020).
  The one collected in Morlaix was the only sample where the B/Ca has significant correlations with temperature proxies (Fig. 8), this is one of the reasons why we decided to reconstruct a detailed age model for this sample. There are fluctuations in B/Ca over time (Fig. 12), but very differently from valuable temperature proxies. We finally explained the correlation of B/Ca with Mg, Sr and Li/Ca in this sample in terms of changes in growth rate that are likely to happen because of the high temperature fluctuations experienced by the alga throughout the year (line 352-356), which make the difference from the other sites. In this paper, we show several results and elaboration from various sampling sites in different ways, with the aim to investigate the growth rate and temperature relationship with B/Ca in *L. corallioides*. In the Mediterranean samples, we did not see any evidence of temperature control over B/Ca. Because of these results we suggest a stronger control of growth rates.

- **Comment:** Abstract/Conclusion – incorporate, or mention, new Li/Mg
  **Response:** Thank you for the suggestion, the change has been made.

- **Comment:** Line29 – "their longevity by indeterminate growth, with no ontogenetic trend", what do you mean by longevity by indeterminate growth? – Ok, I see this is answered in line 30, perhaps split the sentence. 'As well as having longevity by indeterminate growth, that is the growth trend…'
  **Response:** OK

- **Comment:** Line 74: Replace 'actually' with ' For instance, more recently'
  **Response:** OK

- **Comment:** Line 92 – 105: Feel like some of the parts feel unconnected, would reword to something like, to ensure that there is flow: "Here, we present the first laser ablation inductively coupled plasma mass spectrometry (LA-ICP-MS) conducted on coralline algae. This technique, which allows high-resolution analysis of a broad range of trace elements in solid-state samples, has been widely used in biogenic carbonates to extract records of seawater temperature, salinity, and water chemistry (Schöne et al., 2005; Corrège, 2006; Hetzinger et al., 2009, 2011; Fietzke et al., 2015; Ragazzola et al., 2020). Measurements were made on the non-geniculate coralline alga *Lithothamnion corallioides* (P. Crouan & H. Crouan) P. Crouan & H. Crouan 1867 which is widely distributed in the Mediterranean Sea and in the north-eastern Atlantic Ocean, from Scotland to Canary Islands (Irvine and Chamberlain, 1999; Wilson et al., 2004; Carro et al., 2014), usually constituting maerl beds (Potin et al., 1990; Foster, 2001; Martin et al., 2006; Savini et al., 2012; Basso et al., 2017). It forms rhodoliths as unattached branches (Basso et al., 2016) with obvious banding in longitudinal sections (Basso, 1995b). These characteristics combine to make this species a suitable model for the measurement of geochemical proxies, comparing different environmental settings. In this paper, we provide the first LA-ICP-MS data of temperature proxies (Mg/Ca, Sr/Ca, Li/Ca) and B/Ca were measured on *L. corallioides* collected from different geographic settings and depths across the Mediterranean Sea and in the Atlantic Ocean. We test the influence of temperature and growth rate on the B/Ca ratio, which could be crucial in assessing the reliability of B/Ca as a proxy of the seawater carbonate system."
  **Response:** The text has been revised according to the suggestions. Note that this is not the first LA-ICP-MS on coralline algae in general, but the first on a coralline alga grown in nature in two different Basins.

- **Comment:** Line 154/Table 1: I note that Table 1 no longer has the time interval of the ORAS5 datasets – so I assume that all of the 11 year intervals no longer go beyond the collection date? As per reviewer 3
  **Response:** Exactly, as mentioned in the response to Reviewer #3, the time interval extracted for Aegadian Isl. has been corrected in the first revised version. The time interval considered are clearly visible in this newly revised version as suggested (Supplement Fig. S3, S4, S5, S6).

- **Comment:** Line 157: "Missing element ratios were calculated as the means of known values" is it not easier to remove the years corresponding with the known? How do you ensure that your gap-filling doesn't introduce error/uncertainty?
  **Response:** We believe that introducing the mean values of the coldest and warmest months is a reasonable approximation. Moreover, the resulting plot shall be easier for the reader. The few gaps have also been highlighted in the plot as asterisks, following what you suggested in your previous comment.

- **Comment:** Line 160-165: If there is a trend in pH changing through time, is it not prudent to use earlier years than the full range 1993-2018? The authors need to plot the DIC, temperature, and pH values through time for each site in a supplementary figure, identifying which periods were used in their analysis.
  **Response:** Very recently, the CMEMS products used for the environmental data reconstruction have been updated. We therefore decided to update our work accordingly. The new references have been indicated in "Data availability". The time collection of our samples goes from 1990 to 2017, therefore using earlier years did not seem a proper solution. As mentioned to the previous reviewers, DIC data are more sparse, and we did not have the same consistency as for temperature data. We therefore decided to choose a reasonable time period, the same for every sampling site for data comparability. Now, as you better see in the supplementary figures you suggested to add (Fig. S3, S4, S5, S6), the range of variations did not change much through time, and the data well characterize the sites. Thanks to the updated version of the DIC data, we were able to extend the extracted period to 1999-2017. For consistency, we revised the period of extraction of pH data as the same (1999-2017). As you can see from the revised Table 1, the values have slightly changed and the differences among sites remained almost the same. We updated temperature data as well, still with very small changes. Figures 7, 10 have also been revised according to the new data.

- **Comment:** Line 170: "This step was helpful in highlighting faint bands and to achieve a more reliable estimate of the algal growth." Identify which, if any, bands were identified via Mg/Ca or whose position was modified by this method.
  **Response:** Mg/Ca data have been used to help identifying the poorly visible bands of the Elba sample while measuring the growth rates. The text has been revised for clarity because we understood that the sentence created misunderstandings.

- **Comment:** Line 172: "Intermediate Mg/Ca values, indeed, would probably correspond to middle seasons" – if long and short cells are produced in cold and warm seasons, and also intermediate are middle seasons, how reliable is the assumption then that each long and short cell can be assumed to be a season allowing for estimate of the year by counting backwards? Why were these measurements excluded from the dataset?
  **Response:** No measurements have been excluded from the dataset. As mentioned in the response above, Mg/Ca results have been used to aid the identification of growth bands in the Elba sample, where, as you see in the Supplement (Fig. S1), the growth bands were very poorly visible. In the other samples, the growth bands have been identified under light microscope as wrote in Materials and Methods. The text has been revised.

- **Comment:** Line 179: "Data from faint bands had been excluded from the dataset." – why? In the Elba image there are no growth bands visible so were all of these excluded? Likewise Aegadian Island. Note – I now see that is the case but this should be mentioned here.
  **Response:** As mentioned before, the text has been revised in sections 2.4 and 2.5. It was not possible to distinguish clearly long and short cells in the Elba sample because of the poor resolution of the growth bands. Therefore, when analysing the elemental data from short and long cells, the Elba sample is missing. As you suggested, this information has been added in section 2.5. Concerning the measurements of growth rates, we directly observed the section under a light microscope, which was much more effective in visualizing the bands rather than the image analysis (see 2.4). Given the poor resolution of the bands in Elba, and since the record of Mg/Ca was relatively continuous, we also used Mg/Ca results to help measure the growth rate in this sample. The text has been revised for clarity.

- **Comment:** Table 1: the caption states: "Data from monthly means extracted by 11 years of ORAS5 reanalysis. pH and DIC in each sampling site are also indicated. The minimum, maximum, mean, and standard deviation values have been measured on the time interval 2019-2020." Why isn't the pH interval included? I.e., 1993-2018. Is it just DIC covered by the interval 2019-2020?
  **Response:** The caption has been revised adding the pH interval as well.

- **Comment:** Figure 6: Why plot the average of the sites long and short cells for your data? In figure 7 the authors plot temperature and corresponding Mg/Li, why then not do this for this figure? I note that in Figure 7 the Mg/Li of Morlaix would cover nearly the full range of the figure 6's y-axis.
  **Response:** We reconstruct the age model for the Morlaix sample only, for the above-mentioned reasons. Nevertheless, Fig. 6 is no less significant. Indeed, as extensively mentioned in the previous responses and in the last revision, the resolution of the laser ablation does not allow us to attribute a single point of analysis to an absolute temperature in a specific time of the year, but rather refer more generically to the cold and warm seasons. In fact, dark and light bands are usually associated to cold and warm periods, and we have used this estimation to create the plots. When plotting the data collected from all the samples (Fig. 6), we refer to the maximum and minimum temperature registered in the site, corresponding respectively to the long and short cells. When plotting the age model of the Morlaix sample, we always refer to the coldest and warmest month of the year, with the respectively dark and light bands (Fig. 7, 11). As mentioned previously, we added a more detailed explanation on the use of temperature data for the purpose of the paper in Materials and Methods, section 2.3 (lines 157-169).

- **Comment:** Figure 8: This should be for all sites and not just Morlaix, I do not get the rationale for showing only a single site.
  **Response:** The correlation was statistically significant only in the Morlaix sample (line 252).

- **Comment:** Line 273-279: The samples have 8, 10, 11 and 11 years of growth, when then (figure 11) was 11 years of temperature compared for all?
  **Response:** The values of growth rates are approximate due to the limit of the method and the complexity of the algal growth behaviour. For the comparability among sites, we decided for a time interval which almost surely included the whole length of the laser transects. Of course, when

reconstructing the age model of the Morlaix samples, the years indicated effectively correspond to the laser spot.

- **Comment:** Figure 11: Why split the data into short and long cells only for the temperature dataset? Why not plot the growth rates and deltaT against the short and long cells?
  **Response:** Short cells conventionally correspond to the minimum temperature registered in the sites; long cells correspond to the maximum temperature (Moberly, 1970; Corrège, 2006; Williams et al., 2014; Caragnano et al., 2014; Ragazzola et al., 2020). We do not have the growth rates for each single band for the reasons explained above. We believe that our use of temperature data is much clearer in the newly revised text; deltaT plotted against short and long cells would not be meaningful.

- **Comment:** Line 301 : "In the sample from Elba the growth bands were not clearly visible, preventing the analyses of trace elements in long and short cells separately" – this shouldn't pop up in the discussion but should be referred to way earlier.
  **Response:** We implemented the Materials and Methods section 2.5.

**References**

Caragnano, A., Basso, D., Jacob, D. E., Storz, D., Rodondi, G., Benzoni, F. and Dutrieux, E.: Coralline red alga *Lithophyllum kotschyanum* f. affine as proxy of climate variability in the Yemen coast, Gulf of Aden (NW Indian Ocean), Geochim. Cosmochim. Acta, 124, 1–17, doi:10.1016/j.gca.2013.09.021, 2014.

Corrège, T.: Sea surface temperature and salinity reconstruction from coral geochemical tracers, Palaeogeogr., Palaeoclimatol., Palaeoecol., 232, 408–428, doi:10.1016/j.palaeo.2005.10.014, 2006.

Moberly, R: Microprobe study of diagenesis in calcareous algae, Sedimentology, 14: 113–123, doi: 10.1111/j.1365-3091.1970.tb00185.x, 1970.

Halfar, J., Zack, T., Kronz, A. and Zachos, J. C.: Growth and high-resolution paleoenvironmental signals of rhodoliths (coralline red algae): A new biogenic archive, J. Geophys. Res., 105: 22,107-22,116, doi: 10.1029/1999jc000128, 2000.

Hetzinger, S., Halfar, J., Kronz, A., Steneck, R., Adey, W. H., Philipp, A. L. and Schöne, B.: High-resolution Mg/Ca ratios in a coralline red alga as a proxy for Bering Sea temperature variations from 1902-1967, Palaois, 24, 406–412, doi:10.2110/palo.2008.p08-116r, 2009.

Ragazzola, F., Caragnano, A., Basso, D., Schmidt, D. N. and Fietzke, J.: Establishing temperate crustose Early Holocene coralline algae as archived for paleoenvironmental reconstructions of the shallow water habitats of the Mediterranean Sea, Paleontology, 63, 155–170, doi:10.1111/pala.12447, 2020.

Williams, B., Halfar, J., Delong, K. L., Hetzinger, S., Steneck, R. S. and Jacob, D. E.: Multi-specimen and multi-site calibration of Aleutian coralline algal Mg/Ca to sea surface temperature, Geochim. Cosmochim. Acta, 139, 190–204, doi:10.1016/j.gca.2014.04.006, 2014.

---

## Author Response (AR3)

FOR THE EDITOR:

Dear A. Mazumdar,

We are replying below to our 3rd round of major revisions. The revision arrived to us on Oct 11, and we were waiting for it since July 14, which is far beyond an acceptable time for a revision, especially if it is the third one.

The first comment of the reviewer was finally clear to us and we were able to prepare some new pictures that will hopefully solve the issue, and improve the manuscript. The other comments refer either to information that is already contained in the manuscript and already explained in a previous response to the reviewer, or minor clarifications. Are we going to wait for another semester to see a significant step of our manuscript towards publication? We do not understand the protraction of this review, despite no significant major problem.

Thank you for your collaboration.

Comment: The authors have responded to the previous review. Having read the responses and re-read the paper I believe that we are at an impasse. I believe that the revisions highlighted would substantially improve the paper, the authors (somewhat) disagree or have argued against making a number of the suggested changes. There are a number of points that I believe that need to be addressed: the paper is about growth vs. temperature and therefore quantification of both must be as accurate as possible. Yet some of the points raised in the previous review remain unaddressed, for instance, "Why is only the Morlaix site chosen for plotting the comparisons between different variables and not the others? The authors need to show for all samples the transect of laser ablation values vs. length. I would envision this as a plot with several panels, first panel elemental ratio vs depth." The authors state its not possible to plot the growth rates, but the authors could show a plot of the element ratios along the transect (i.e., identifying also where samples were taken) and the rough position of the growth bands, no?

Response: We have created the new figures requested, in Supplements (Figure S1, S2, S3).

Comment: Or the response to asking why the number of years of growth for calculating mm/yr uses the 'exact' number of years (section 3.4) however when computing temperature a fixed number of 11 years is used. In responses to this question the authors state that "The values of growth rates are approximate due to the limit of the method and the complexity of the algal growth behaviour. For the comparability among sites, we decided for a time interval which almost surely included the whole length of the laser transects". The authors argue (in response to a different question) that "the resolution of the laser ablation does not allow us to attribute a single point of analysis to an absolute temperature in a specific time of the year, but rather refer more generically to the cold and warm seasons."

Response: Please mind that there is no correlation between the time interval of extraction and the number of years counted when measuring the growth rates. Following a well-established procedure for the growth rate measurements, we divided the length of the laser transect by the number of annual growth bands crossed (see Section 2.4). This calculation results in the value of mm of growth per year. The time interval of temperature extraction was set to 11 years before sample collection, in every site, and is unrelated to the previous calculation. We revised the manuscript (line 154) to be more explicit about this point.

Comment: Or why is figure 8 not plotting all sites elemental ratios - I ask because Morlaix has a [weak] correlation between B/Ca and Mg/Ca. It would be interesting to know whether this is found among all sites or just one, especially if you consider that Elba you use Mg/Ca to make an inference of growth to compare against B/Ca.

Response: Figure 8 shows the only significant correlation found (see line 254 and responses to previous reviews). Relationships between B/Ca and temperature proxies were calculated for each sample as wrote in the manuscript since the original version, founding no significant correlations (lines 254-255). Therefore, this information is already present, as we commented also in the previous review. Still, the presence or absence of a correlation between Mg/Ca and B/Ca is unrelated to the use of Mg/Ca variations in supporting the distinction between short and long cells. Mg/Ca is a well-established temperature proxy, as also confirmed by our results.

Comment: As such many of the previous questions stand I would therefore still consider it major revisions.

Response: Although we do not see the need for this revision to be major, we have hopefully clarified the points and further implemented the manuscript. We believe that our work, significantly improved by the revisions, includes all the information we could get from our samples concerning growth rate and temperature, in support to our conclusions.

---

## Author Response (AR4)

In response to Referee #4:

the supposed weaknesses underlined in the comments have been addressed as explained in the following lines (red colour).

- Line 24: When selecting papers, cite only the earliest papers identifying these global changes and the most recent ones like advanced IPCC reports.

We modified the citations as suggested.

Lines 40 - 57: The authors could be much more synthetic, summarizing this in a few lines and especially referring only to review articles (Foster, De Carlo, etc.) as well as the latest advances on the reconstruction technique of carbonate chemistry...

The section was shortened, and the review articles have been added in the revised version. The discussion about "the latest advances on the reconstruction technique of carbonate chemistry" is beyond the aim of this paper, which is focused on the investigation of possible temperature and growth rate effects on B/Ca.

Line 57: Not so recent ....

The sentence has been removed and the introduction has been synthetized.

Lines 59-61: what is the interest of this paragraph for this study ?

The idea was to provide an overview of the mechanisms of B incorporation in marine carbonates, but the removal of this sentence does not hamper the significance of the introduction, therefore we removed it.

Introduction - Rather than redoing a history on boron, from my point of view the introduction should have been oriented on the response of the SST and carbonate chemistry proxies according to the type of materials studied, calcite or aragonite and finally yes the little knowledge concerning the bio-carbonates in Mg-Calcite

This would surely be an interesting topic to discuss in another paper, but what you proposed would not be significant for this work, which is specifically devoted to Mg-calcite coralline algae and their incorporation of boron.

Lines 81-82: What is this debate?

The two previous sentences refer to papers with opposite conclusions about the controls on B incorporation (Donald et al., 2017 and Anagnostou et al., 2019). The debate clearly refers to this.

Can we really use Mg-calcite organisms ? Such questions have to be developed in the introduction

The answer will come after grouping the results of several researchers working on different groups. It is exactly the issue addressed in this paper, limitedly to one species of coralline algae, in order to exclude possible odd comparisons coming from species-specific vital effects.

Line 139: in general the authors have to be more precise .... Which laser ???? Line 145: Not clear. higher ???? How much ???? or lower .. please indicate the analytical uncertainties for each isotope/element analysed

The producer has been added, but the characteristics of the instrument were already given. More details about the instrument precision have been given as suggested.

Line 201: the authors have to be less qualitative... Please be more quantitative

A previous revision was asking for eliminating the numbers from this section, because they are quantitatively presented in the cited Table 1. We therefore already provided all the quantitative results.

Line 318: Not only... Geographical locations can play a role too on delta T

We modified the text accordingly.

Line 323: But how much is reliable ?

We provided data and treated them in order to test their statistical significance. Our data are reliable, "how much" is defined by statistics. See section 3.2.

Lines 328-330: But why the authors did not the tests and calculations of the interest of a multi-proxy approach ??? Need to be develop in this study

We tested the 3 proxies mentioned, and the results are presented in Section 3.2. No modification was needed here.

Line 350 and along the discussion: in general many assumptions in this study without solid arguments

No precise comment or criticism is expressed here, providing no effective suggestion for implementation.

Discussion: Why do the authors never discuss the potential role of the organism in up-regulating the carbonate chemistry of their internal calcifying fluid (here $CO_3^{2-}$) and consequently the growth parameters, here the linear extension... I am really not convinced by the discussion presented here and the fact that the authors ignore all the recent works on these geochemical processes.

All the papers related to coralline algae (and not only) have been considered in our discussion. The physiology of a possible up-regulation in the calcifying fluids in coralline algae is a topic for further research and different approaches, that are beyond the scope of this contribution.

---

## Author Response (AR5)

Comment:

1. Please see the comment: Introduction - Rather than redoing a history on boron, from my point of view the introduction should have been oriented on the response of the SST and carbonate chemistry proxies according to the type of materials studied, calcite or aragonite, and finally yes the little knowledge concerning the bio-carbonates in Mg-Calcite

Response:

We modified the introduction about boron incorporation in marine carbonates, including only the relevant information for the aim of the study. Temperature proxies in Mg-calcites have a long and robust record in literature, and all the significant references on the subject have been included in the introduction (from line 31, and modified text from line 42 in the paper with track changes). We also included a brief statement about the importance of the mineralogical control over biological up-regulation, highlighting differences between aragonite and calcite (from line 82 of the revised text with track changes).

Comment:

2. Can we really use Mg-calcite organisms? Such questions have to be developed in the introduction

Response:

The value of Mg-calcites and especially coralline algae as recorders of past climate has been discussed since decades (Chave and Wheeler, 1965; Moberly, 1968), and evidences proved their suitability as paleoclimate archives. We further stressed their meaningfulness in the introduction (lines 42-46).

3. Discussion: Why do the authors never discuss the potential role of the organism in up-regulating the carbonate chemistry of their internal calcifying fluid (here CO32-) and consequently the growth parameters, here the linear extension... I am really not convinced by the discussion presented here and the fact that the authors ignore all the recent works on these geochemical processes.

Response:

We discussed about the up-regulation of the calcifying fluid ($pH_{cf}$) in corals and coralline algae at lines 82-82 and from line 386 of the revised text with track changes. We therefore cited recent works on $pH_{cf}$ accordingly, which suggest a species-specific control over $pH_{cf}$ at different ambient pH. Despite being an interesting topic, the elevation of pH at the site of calcification would not be determinant for the variations observed in our data. Indeed, we examined a single species, avoiding the problem of species-specific differences in up-regulations. Moreover, all samples have been collected in normal seawater pH, without significant variations among sampling sites. Therefore, we would not expect differences in up-regulations that could control the measured B/Ca or the growth rates. We also revised some points in the text throughout, for clarity.

References

Chave, K.E., and Wheeler, B.D., Jr., 1965, Mineralogic changes during growth in the red alga

Clathromorphum compactum: Science, v. 147, p. 621, doi: 10.1126/science.147.3658.621.

Moberly, R., Jr., 1968, Composition of magnesian calcites of algae and pelecypods by electron

microprobe analysis: Sedimentology, v. 11, p. 61–82, doi: 10.1111/j.1365–3091.1968.tb00841.x.